# Learning to Guide Random Search

**Ozan Sener**
Intel Labs

**Vladlen Koltun**
Intel Labs

## Abstract

We are interested in derivative-free optimization of high-dimensional functions. The sample complexity of existing methods is high and depends on problem dimensionality, unlike the dimensionality-independent rates of first-order methods. The recent success of deep learning suggests that many datasets lie on low-dimensional manifolds that can be represented by deep nonlinear models. We therefore consider derivative-free optimization of a high-dimensional function that lies on a latent low-dimensional manifold. We develop an online learning approach that learns this manifold while performing the optimization. In other words, we jointly learn the manifold and optimize the function. Our analysis suggests that the presented method significantly reduces sample complexity. We empirically evaluate the method on continuous optimization benchmarks and high-dimensional continuous control problems. Our method achieves significantly lower sample complexity than Augmented Random Search, Bayesian optimization, covariance matrix adaptation (CMA-ES), and other derivative-free optimization algorithms.

## 1 Introduction

A typical approach to machine learning problems is to define an objective function and optimize it over a dataset. First-order optimization methods are widely used for this purpose since they can scale to high-dimensional problems and their convergence rates are independent of problem dimensionality in most cases. However, gradients are not available in many important settings such as control, black-box optimization, and interactive learning with humans in the loop. Derivative-free optimization (DFO) can be used to tackle such problems. The challenge is that the sample complexity of DFO scales poorly with the problem dimensionality. The design of DFO methods that solve high-dimensional problems with low sample complexity is a major open problem.

The success of deep learning methods suggests that high-dimensional data that arises in real-world settings can commonly be represented in low-dimensional spaces via learned nonlinear features. In other words, while the problems of interest are high-dimensional, the data typically lies on low-dimensional manifolds. If we could perform the optimization directly in the manifold instead of the full space, intuition suggests that we could reduce the sample complexity of DFO methods since their convergence rates are generally a function of the problem dimensionality (Nesterov & Spokoiny, 2017; Dvurechensky et al., 2018). In this paper, we focus on high-dimensional data distributions that are drawn from low-dimensional manifolds. Since the manifold is typically not known prior to the optimization, we pose the following question. *Can we develop an adaptive derivative-free optimization algorithm that learns the manifold in an online fashion while performing the optimization?*

There exist DFO methods that aim to identify a low-dimensional search space (Maheswaranathan et al., 2019; Choromanski et al., 2019). However, they are limited to linear subspaces. In contrast, we propose to use expressive nonlinear models (specifically, neural networks) to represent the manifold. Our approach not only increases expressivity but also enables utilization of domain knowledge concerning the geometry of the problem. For example, if the function of interest is known to be translation invariant, convolutional networks can be used to represent the underlying manifold structure. On the other hand, the high expressive power and flexibility brings challenges. Our approach requires solving for the parameters of the nonlinear manifold at each iteration of the optimization. To address this, we develop an efficient online method that learns the underlying manifold while the function is being optimized.

We specifically consider random search methods and extend them to the nonlinear manifold learning setting. Random search methods choose a set of random directions and perform perturbations to the current iterate in these directions. Differences of the function values computed at perturbed points are used to compute an estimator for the gradient of the function. We first extend this to random search over a known manifold and show that sampling directions in the tangent space of the manifold provides a similar estimate. We then propose an online learning method that estimates this manifold while jointly performing the optimization. We theoretically analyze sample complexity and show that our method reduces it. We conduct extensive experiments on continuous control problems, continuous optimization benchmarks, and gradient-free optimization of an airfoil. The results indicate that our method significantly outperforms state-of-the-art derivative-free optimization algorithms from multiple research communities.

## 2 PRELIMINARIES

We are interested in high-dimensional stochastic optimization problems of the form

$$\min_{\mathbf{x} \in \mathbb{R}^d} f(\mathbf{x}) = \mathbb{E}_\xi[F(\mathbf{x}, \xi)], \tag{1}$$

where $\mathbf{x}$ is the optimization variable and $f : \mathbb{R}^d \to \mathbb{R}$ is the function of interest, which is defined as expectation over a noise variable $\xi$. We assume that the stochastic function is bounded ($|F(\mathbf{x}, \xi)| \leq \Omega$), $L$-Lipschitz, and $\mu$-smooth[1] with respect to $\mathbf{x}$ for all $\xi$, and has uniformly bounded variance ($\mathbb{E}_\xi[(F(\mathbf{x}, \xi) - f(\mathbf{x}))^2] \leq V_F$). In DFO, we have no access to the gradients. Instead, we only have zeroth-order access by evaluating the function $F$ (i.e. sampling $F(\mathbf{x}, \xi)$ for the input $\mathbf{x}$).

We are specifically interested in random search methods in which an estimate of the gradient is computed using function evaluations at points randomly sampled around the current iterate. Before we formalize this, we introduce some definitions. Denote the $d$-dimensional unit sphere and unit ball by $\mathbb{S}^{d-1}$ and $\mathbb{B}^d$, respectively. We define a *smoothed* function following Flaxman et al. (2005). For a function $f : \mathbb{R}^d \to \mathbb{R}$, its $\delta$-smoothed version is $\hat{f}(\mathbf{x}) = \mathbb{E}_{\mathbf{v} \sim \mathbb{B}^d}[f(\mathbf{x} + \delta \mathbf{v})]$. The main workhorse of random search is the following result by Flaxman et al. (2005). Let $\mathbf{s}$ be a random vector sampled from the uniform distribution over $\mathbb{S}^{d-1}$. Then $f(\mathbf{x} + \delta \mathbf{s})\mathbf{s}$ is an unbiased estimate of the gradient of the smoothed function:

$$\mathbb{E}_{\xi, \mathbf{s} \in \mathbb{S}^{d-1}}[F(\mathbf{x} + \delta\mathbf{s}, \xi)\mathbf{s}] = \frac{\delta}{d}\nabla_{\mathbf{x}}\hat{f}(\mathbf{x}). \tag{2}$$

We use antithetic samples since this is known to decrease variance (Kroese et al., 2013) and define the final gradient estimator as $y(\mathbf{x}, \mathbf{s}) = (F(\mathbf{x} + \delta\mathbf{s}, \xi) - F(\mathbf{x} - \delta\mathbf{s}, \xi))\mathbf{s}$. Extending (2) to the antithetic case, $\mathbb{E}_{\xi, \mathbf{s} \in \mathbb{S}^{d-1}}[y(\mathbf{x}, \mathbf{s})] = \frac{2\delta}{d}\nabla_{\mathbf{x}}\hat{f}(\mathbf{x})$.

A simple way to optimize the function of interest is to use the gradient estimate in stochastic gradient descent (SGD), as summarized in Algorithm 1. This method has been analyzed in various forms and its convergence is characterized well for nonconvex smooth functions. We restate the convergence rate and defer the constants and proof to Appendix A.2.

**Proposition 1** (Flaxman et al., 2005; Vemula et al., 2019). *Let $f(\mathbf{x})$ be differentiable, $L$-Lipschitz, and $\mu$-smooth. Consider running random search (Algorithm 1) for $T$ steps. Let $k = 1$ for simplicity. Then*

$$\frac{1}{T}\sum_{t=1}^{T} \mathbb{E}\|\nabla_{\mathbf{x}}f(\mathbf{x}^t)\|_2^2 \leq \mathcal{O}\left(T^{-\frac{1}{2}}d + T^{-\frac{1}{3}}d^{\frac{2}{3}}\right).$$

## 3 ONLINE LEARNING TO GUIDE RANDOM SEARCH

Proposition 1 implies that the sample complexity of random search scales linearly with the dimensionality. This dependency is problematic when the function of interest is high-dimensional. We argue that in many practical problems, the function of interest lies on a low-dimensional nonlinear

---

[1] $L$-Lipschitz and $\mu$-smooth: $|f(\mathbf{x}) - f(\mathbf{y})| \leq L\|\mathbf{x} - \mathbf{y}\|_2$ and $\|\nabla_{\mathbf{x}}f(\mathbf{x}) - \nabla_{\mathbf{x}}f(\mathbf{y})\|_2 \leq \mu\|\mathbf{x} - \mathbf{y}\|_2 \quad \forall_{\mathbf{x}, \mathbf{y}}$

---

**Algorithm 1** Random Search

1: **for** $t = 1$ **to** $T$ **do**
2:     $\mathbf{g}^t, \_ = \text{GRADEST}(\mathbf{x}^t, \delta)$
3:     $\mathbf{x}^{t+1} = \mathbf{x}^t - \alpha \mathbf{g}^t$
4: **end for**

---

1: **procedure** GRADEST($\mathbf{x}, \delta$)
2:     **Sample:** $\mathbf{s}_1, \ldots, \mathbf{s}_k \sim \mathbb{S}^{d-1}$
3:     **Query:** $y_i = f(\mathbf{x} + \delta \mathbf{s}_i) - f(\mathbf{x} - \delta \mathbf{s}_i)$
4:     **Estimator:** $\mathbf{g} = \frac{d}{2\delta} \sum_{i=1}^{k} y_i \mathbf{s}_i$
5:     **return** $\mathbf{g}, \{\mathbf{s}_i\}_{i \in [k]}$
6: **end procedure**

---

**Algorithm 2** Manifold Random Search

1: **for** $t = 1$ **to** $T$ **do**
2:     $\mathbf{g}^t, \_ = \text{MANIFOLDGRADEST}(\mathbf{x}^t, \mathbf{J}(\mathbf{x}^t; \theta^\star))$
3:     $\mathbf{x}^{t+1} = \mathbf{x}^t - \alpha \mathbf{g}^t$
4: **end for**

---

1: **procedure** MANIFOLDGRADEST($\mathbf{x}, \theta, \delta$)
2:     **Normalize:** $\mathbf{J}_q = \text{GramSchmidt}(\mathbf{J})$
3:     **Sample:** $\mathbf{s}_1, \ldots, \mathbf{s}_k \sim \mathbb{S}^{n-1}$
4:     **Query:** $y_i = f(\mathbf{x} + \delta \mathbf{J}_q \mathbf{s}_i) - f(\mathbf{x} - \delta \mathbf{J}_q \mathbf{s}_i)$
5:     **Estimator:** $\mathbf{g} = \frac{n}{2\delta} \sum_{i=1}^{k} y_i \mathbf{J}_q \mathbf{s}_i$
6:     **return** $\mathbf{g}, \{\mathbf{J}_q \mathbf{s}_i\}_{i \in [k]}$
7: **end procedure**

---

manifold. This structural assumption will allow us to significantly reduce the sample complexity of random search, without knowing the manifold a priori.

Assume that the function of interest is defined on an $n$-dimensional manifold ($n \ll d$) and this manifold can be defined via a nonlinear parametric family (e.g. a neural network). Formally, we are interested in derivative-free optimization of functions with the following properties:

- **Smoothness:** $F(\cdot, \xi) : \mathbb{R}^d \to \mathbb{R}$ is $\mu$-smooth and $L$-Lipschitz for all $\xi$.
- **Manifold:** $F(\cdot, \xi)$ is defined on an $n$-dimensional manifold $\mathcal{M}$ for all $\xi$.
- **Representability:** The manifold $\mathcal{M}$ and the function of interest can be represented using parametrized function classes $r(\cdot; \theta)$ and $g(\cdot; \psi)$. Formally, given $\xi$, there exist $\theta^\star$ and $\psi^\star$ such that $F(\mathbf{x}, \xi) = g(r(\mathbf{x}; \theta^\star); \psi^\star) \quad \forall \mathbf{x} \in \mathbb{R}^d$.

We will first consider an idealized setting where the manifold is already known (i.e. we know $\theta^\star$). Then we will extend the developed method to the practical setting where the manifold is not known in advance and must be estimated with no prior knowledge as the optimization progresses.

### 3.1 WARM-UP: RANDOM SEARCH OVER A KNOWN MANIFOLD

If the manifold is known a priori, we can perform random search directly over the manifold instead of the full space. Consider the chain rule applied to $g(r(\mathbf{x}; \theta); \psi)$ as $\nabla_\mathbf{x} f(\mathbf{x}) = \mathbf{J}(\mathbf{x}; \theta^\star) \nabla_\mathbf{r} g(\mathbf{r}; \psi)$, where $\mathbf{J}(\mathbf{x}; \theta^\star) = \partial r(\mathbf{x}; \theta^\star) / \partial \mathbf{x}$ and $\mathbf{r} = r(\mathbf{x}, \theta^\star)$. The gradient of the function of interest lies in the column space of the Jacobian of the parametric family. In light of this result, we can perform random search in the column space of the Jacobian, which has lower dimensionality than the full space.

For numerical stability, we will first orthonormalize the Jacobian using the Gram-Schmidt procedure, and perform the search in the column space of this orthonormal matrix since it spans the same space. We denote the orthonormalized version of $\mathbf{J}(\mathbf{x}; \theta^\star)$ by $\mathbf{J}_q(\mathbf{x}; \theta^\star)$.

In order to perform random search, we sample an $n$-dimensional vector uniformly ($\tilde{\mathbf{s}} \sim \mathbb{S}^{n-1}$) and lift it to the input space via $\mathbf{J}_q(\mathbf{x}; \theta^\star)\tilde{\mathbf{s}}$. As a consequence of the manifold Stokes' theorem, using the lifted vector as a random direction gives an unbiased estimate of the gradient of the smoothed function as

$$\mathbb{E}_{\xi, \mathbf{s} \sim \mathbb{S}^{n-1}}[y(\mathbf{x}, \mathbf{J}_q(\mathbf{x}; \theta^\star)\tilde{\mathbf{s}})] = \frac{2\delta}{n} \nabla_\mathbf{x} \tilde{f}_{\theta^\star}(\mathbf{x}), \tag{3}$$

where the smoothed function is defined as $\tilde{f}_{\theta^\star}(\mathbf{x}) = \mathbb{E}_{\tilde{\mathbf{v}} \sim \mathbb{B}^n}[f(\mathbf{x} + \delta \mathbf{J}_q(\mathbf{x}; \theta^\star)\tilde{\mathbf{v}})]$. We show this result as Lemma 1 in Appendix A.1. We use the resulting gradient estimate in SGD. The following proposition summarizes the sample complexity of this method. The constants and the proof are given in Appendix A.2.

**Proposition 2.** *Let $f(\mathbf{x})$ be differentiable, $L$-Lipschitz, and $\mu$-smooth. Consider running manifold random search (Algorithm 2) for $T$ steps. Let $k = 1$ for simplicity. Then*

$$\frac{1}{T} \sum_{t=1}^{T} \mathbb{E}\|\nabla_\mathbf{x} f(\mathbf{x}^t)\|_2^2 \leq \mathcal{O}\left(T^{-\frac{1}{2}} n + T^{-\frac{1}{3}} n^{\frac{2}{3}}\right).$$

## 3.2 JOINT OPTIMIZATION AND MANIFOLD LEARNING

When $n \ll d$, the reduction in the sample complexity of random search (summarized in Proposition 2) is significant. However, the setting of Algorithm 2 and Proposition 2 is impractical since the manifold is generally not known a priori. We thus propose to minimize the function and learn the manifold jointly. In other words, we start with an initial guess of the parameters and solve for them at each iteration using all function evaluations that have been performed so far.

Our major objective is to improve the sample efficiency of random search. Hence, minimizing the sample complexity with respect to manifold parameters is an intuitive way to approach the problem. We analyze the sample complexity of SGD using biased gradients in Appendix A.3.1 and show the following informal result. Consider running manifold random search with a sequence of manifold parameters $\theta^1, \psi^1, \ldots, \theta^T, \psi^T$ for $T$ steps. Then the additional suboptimality caused by biased gradients, defined as $\text{SUBOPTIMALITY} = \frac{1}{T} \sum_{t=1}^{T} \mathbb{E}[\nabla_\mathbf{x} f(\mathbf{x})]$, is bounded as follows:

$$\text{SUBOPTIMALITY}(\theta^t, \psi^t) \leq \text{SUBOPTIMALITY}(\theta^\star, \psi^\star) + \frac{\Omega}{T} \sum_{t=1}^{T} \|\nabla_\mathbf{x} \tilde{f}_{\theta^\star}(\mathbf{x}^t) - \nabla_\mathbf{x} g(r(\mathbf{x}^t; \theta^t); \psi^t)\|_2, \quad (4)$$

where $\text{SUBOPTIMALITY}(\theta^\star, \psi^\star)$ is the suboptimality of the oracle case (Algorithm 2). Our aim is to minimize the additional suboptimality with respect to $\theta^t$ and $\psi^t$. However, we do not have access to $\nabla_\mathbf{x} f(\mathbf{x})$ since we are in a derivative-free setting. Hence we cannot directly minimize (4).

At each iteration, we observe $y(\mathbf{x}^t, \mathbf{s}^t)$. Moreover, $y(\mathbf{x}^t, \mathbf{s}^t) = 2\delta \mathbf{s}^{t\intercal} \nabla_\mathbf{x} \tilde{F}(\mathbf{x}^t, \xi) + \mathcal{O}(\delta^2)$, due to the smoothness. Since we observe the projection of the gradient onto the chosen directions, we minimize the projection of (4) onto these directions. Formally, we define our one-step loss as

$$\mathcal{L}(\mathbf{x}^t, \mathbf{s}^t, \theta^t, \psi^t) = \left( \frac{y(\mathbf{x}^t, \mathbf{s}^t)}{2\delta} - \mathbf{s}^{t\intercal} \nabla_\mathbf{x} g(r(\mathbf{x}^t; \theta^t); \psi^t) \right)^2. \quad (5)$$

We use the Follow the Regularized Leader (FTRL) algorithm (Hazan, 2016; Shalev-Shwartz, 2012) to minimize the aforementioned loss function and learn the manifold parameters:

$$\theta^{t+1}, \psi^{t+1} = \arg\min_{\theta, \psi} \sum_{i=1}^{t} \mathcal{L}(\mathbf{x}^i, \mathbf{s}^i, \theta, \psi) + \lambda \mathcal{R}(\theta, \psi), \quad (6)$$

where the regularizer $\mathcal{R}(\theta, \psi) = \|\nabla_\mathbf{x} g(r(\mathbf{x}^t; \theta^t); \psi^t) - \nabla_\mathbf{x} g(r(\mathbf{x}^t; \theta); \psi)\|_2$ is a temporal smoothness term that penalizes sudden changes in the gradient estimates.

Algorithm 3 summarizes our algorithm. We add exploration by sampling a mix of directions from the manifold and the full space. In each iteration, we sample directions and produce two gradient estimates $\mathbf{g}_m, \mathbf{g}_e$ using the samples from the tangent space and the full space, respectively. We mix them to obtain the final estimate $\mathbf{g} = (1 - \beta)\mathbf{g}_m + \beta\mathbf{g}_e$. We discuss the implementation details of the FTRL step in Section 4. In our theoretical analysis, we assume that (6) can be solved optimally. Although this is a strong assumption, experimental results suggest that neural networks can easily fit any training data (Zhang et al., 2017). Our experiments also support this observation.

Theorem 1 states our main result concerning the sample complexity of our method. As expected, the sample complexity includes both the input dimensionality $d$ and the manifold dimensionality $n$. On the other hand, the sample complexity only depends on $n\sqrt{d}$ rather than $d$. Thus our method significantly decreases sample complexity when $n \ll d$.

**Theorem 1.** *Let $f(\mathbf{x})$ be bounded, $L$-Lipschitz, and $\mu$-smooth. Consider running learned manifold random search (Algorithm 3) for $T$ steps. Let $k_e = 1$ and $k_m = 1$ for simplicity. Then*

$$\frac{1}{T} \sum_{i=1}^{T} \mathbb{E}\|\nabla_\mathbf{x} f(\mathbf{x}^t)\|_2^2 \leq \mathcal{O}\left( d^{\frac{1}{2}} T^{-1} + (d^{\frac{1}{2}} + n + nd^{\frac{1}{2}}) T^{-\frac{1}{2}} + (n^{\frac{2}{3}} + d^{\frac{1}{2}} n^{\frac{2}{3}}) T^{-\frac{1}{3}} \right).$$

*Proof sketch.* We provide a short proof sketch here and defer the detailed proof and constants to Appendix A.3. We start by analyzing SGD with bias. The additional suboptimality of using $\theta^t, \psi^t$ instead of $\theta^\star, \psi^\star$ can be bounded by (4).

The empirical loss we minimize is the projection of (4) onto randomly chosen directions. Next, we show that the expectation of the empirical loss is (4) when the directions are chosen uniformly at random from the unit sphere:

$$\mathbb{E}_{\mathbf{s}^t \in \mathbb{S}^{d-1}} \left[ \mathcal{L}(\mathbf{x}^t, \mathbf{s}^t, \theta^t, \psi^t) \right] = \frac{1}{dT} \sum_{t=1}^{T} \|\nabla_{\mathbf{x}} \tilde{f}_{\theta^\star}(\mathbf{x}^t) - \nabla_{\mathbf{x}} g(r(\mathbf{x}^t; \theta^t); \psi^t)\|_2. \tag{7}$$

A crucial argument in our analysis is the concentration of the empirical loss around its expectation. In order to study this concentration, we use Freedman's inequality (Freedman, 1975), inspired by the analysis of generalization in online learning by Kakade & Tewari (2009). Our analysis bounds the difference $\left| \mathbb{E}_{\mathbf{s}^t \in \mathbb{S}^{d-1}} \left[ \sum_{t=1}^{T} \mathcal{L}^t \right] - \sum_{t=1}^{T} \mathcal{L}^t \right|$, where $\mathcal{L}^t = \mathcal{L}(\mathbf{x}^t, \mathbf{s}^t, \theta^t, \psi^t)$.

Next, we use the FTL-BTL Lemma (Kalai & Vempala, 2005) to analyze the empirical loss $\sum_{t=1}^{T} \mathcal{L}^t$. We bound the empirical loss in terms of the distances between the iterates $\sum_{t=1}^{T} \|\mathbf{x}^{t+1} - \mathbf{x}^t\|_2$. Such a bound would not be useful in an adversarial setting since the adversary chooses $\mathbf{x}^t$, but we set appropriate step sizes, which yield sufficiently small steps and facilitate convergence.

Our analysis of learning requires the directions in (7) to be sampled from a unit sphere. On the other hand, our optimization method requires directions to be chosen from the tangent space of the manifold. We mix exploration (directions sampled from $\mathbb{S}^{d-1}$) and exploitation (directions sampled from the tangent space of the manifold) to address this mismatch. We show that mixing weight $\beta = 1/d$ yields both fast optimization and no-regret learning. Finally, we combine the analyses of empirical loss, concentration, and SGD to obtain the statement of the theorem. $\qquad\square$

## 4   IMPLEMENTATION DETAILS AND LIMITATIONS

We summarize important details here and elaborate further in Appendix B. A full implementation is available at `https://github.com/intel-isl/LMRS`.

**Parametric family.** We use multilayer perceptrons with ReLU nonlinearities to define $g$ and $r$. We initialize our models with standard normal distributions. Our method thus starts with random search at initialization and transitions to manifold random search as the learning progresses.

**Solving FTRL.** Results on training deep networks suggest that local SGD-based methods perform well. We thus use SGD with momentum as a solver for FTRL in (6). We do not solve each learning problem from scratch but initialize with the previous solution. Since this process may be vulnerable to local optima, we fully solve (6) from scratch for every 100[th] iteration of the method.

**Computational complexity.** Our method increases the amount of computation since we need to learn a model while performing the optimization. However, in DFO, the major computational bottleneck is typically the function evaluation. When efficiently implemented on a GPU, the time spent on learning the manifold is negligible in comparison to function evaluations.

**Parallelization.** Random search is highly parallelizable since directions can be processed independently. Communication costs include i) sending the current iterate to workers, ii) sending directions to each corresponding worker, and iii) workers sending the function values back. When the directions are chosen independently, they can be indicated to each worker via a single integer by first creating a shared noise table in preprocessing. For a $d$-dimensional problem with $k$ random directions, these costs are $d$, $k$, and $k$, respectively. The total communication cost is therefore $d + 2k$. In our method, each worker also needs a copy of the Jacobian, resulting in a communication cost of $d + 2k + kd$. Hence our method increases communication cost from $d + 2k$ to $d + 2k + kd$.

---

**Algorithm 3** Learned Manifold Random Search (LMRS)

1: **for** $t = 1$ **to** $T$ **do**
2: $\quad \mathbf{g}_e^t, \mathbf{S}_e^t = \text{GRADEST}(\mathbf{x}^t, \delta)$
3: $\quad \mathbf{g}_m^t, \mathbf{S}_m^t = \text{MANIFOLDGRADEST}(\mathbf{x}^t, \mathbf{J}(\mathbf{x}^t; \theta^t))$
4: $\quad \mathbf{g}^t = \frac{\beta k_e}{k_e + k_m} \mathbf{g}_e^t + \frac{(1-\beta) k_m}{k_e + k_m} \mathbf{g}_m^t$
5: $\quad \mathbf{x}^{t+1} = \mathbf{x}^t - \alpha \mathbf{g}^t$
6: $\quad \theta^{t+1}, \psi^{t+1} = \arg\min_{\theta, \psi} \sum_{i=1}^{t} \mathcal{L}(\mathbf{x}^i, \mathbf{S}_{e,m}^i, \theta, \psi) + \lambda \mathcal{R}(\theta, \psi)$
7: **end for**

See Algorithms 1 & 2 for definitions of GRADEST and MANIFOLDGRADEST.

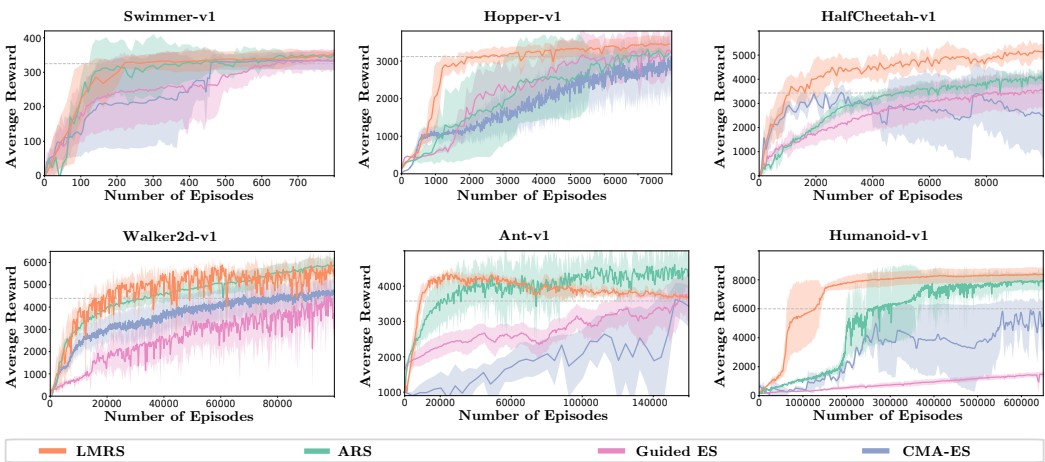

Figure 1: Average reward vs. number of episodes for MuJoCo locomotion tasks. In each condition, we perform 5 runs with different random seeds. Shaded areas represent 1 standard deviation. The grey horizontal line indicates the prescribed threshold at which the task is considered 'solved'.

## 5 EXPERIMENTS

We empirically evaluate the presented method (referred to as Learned Manifold Random Search (LMRS)) on the following sets of problems. i) We use the MuJoCo simulator (Todorov et al., 2012) to evaluate our method on high-dimensional control problems. ii) We use 46 single-objective unconstrained functions from the Pagmo suite of continuous optimization benchmarks (Biscani et al., 2019). iii) We use the XFoil simulator (Drela, 1989) to benchmark gradient-free optimization of an airfoil.

We consider the following baselines. i) Augmented Random Search (ARS): Random search with all the augmentations from Mania et al. (2018). ii) Guided ES (Maheswaranathan et al., 2019): A method to guide random search by adapting the covariance matrix. iii) CMA-ES (Hansen, 2016): Adaptive derivative-free optimization based on evolutionary search. iv) REMBO (Wang et al., 2016): A Bayesian optimization method which uses random embeddings in order to scale to high-dimensional problems. Although CMA-ES and REMBO are not based on random search, we include them for the sake of completeness. Additional implementation details are provided in Appendix B.

### 5.1 LEARNING CONTINUOUS CONTROL

Following the setup of Mania et al. (2018), we use random search to learn control of highly articulated systems. The MuJoCo locomotion suite (Todorov et al., 2012) includes six problems of varying difficulty. We evaluate our method and the baselines on all of them. We use linear policies and include all the tricks (whitening the observation space and scaling the step size using the variance of the rewards) from Mania et al. (2018). We report average reward over five random experiments versus the number of episodes (i.e. number of function evaluations) in Figure 1. We also report the average number of episodes required to reach the prescribed reward threshold at which the task is considered 'solved' in Table 1. We include proximal policy optimization (PPO) (Schulman et al., 2017; Hill et al., 2018) for reference. Note that our results are slightly different from the numbers reported by Mania et al. (2018) as we use 5 random seeds instead of 3.

The results suggest that our method improves upon ARS in all environments. Our method also outperforms all other baselines. The improvement is particularly significant for high-dimensional problems such as Humanoid. Our method is at least twice as efficient as ARS in all environments except Swimmer, which is the only low-dimensional problem in the suite. Interestingly, Guided-ES fails to solve the Humanoid task, which we think is due to biased gradient estimation. Furthermore, CMA-ES performs similarly to ARS. These results suggest that a challenging task like Humanoid is out of reach for heuristics like local adaptation of the covariance matrix due to high stochasticity and nonconvexity.

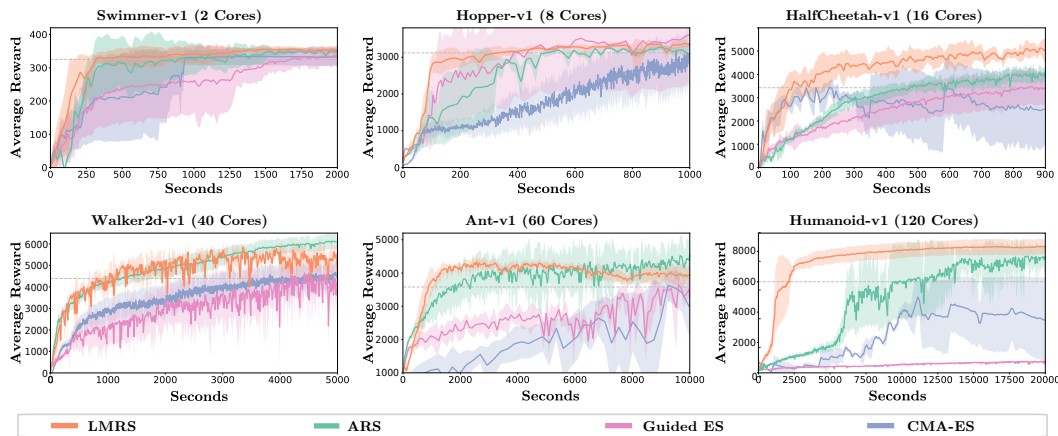

Figure 2: Average reward vs. wall-clock time for MuJoCo locomotion tasks. In each condition, we perform 5 runs with different random seeds. Shaded areas represent 1 standard deviation. The grey horizontal line indicates the prescribed threshold at which the task is considered 'solved'. Measurements are performed on Intel Xeon E7-8890 v3 processors and Nvidia GeForce RTX 2080 Ti GPUs. We list the number of cores used for each experiment.

REMBO only solves the Swimmer task and fails to solve others. We believe this is due to the fact that these continuous control problems have no global structure and are highly nonsmooth. The number of possible sets of contacts with the environment is combinatorial in the number of joints, and each contact configuration yields a distinct reward surface. This contradicts the global structure assumption in Bayesian optimization.

**Wall-clock time analysis.** Our method performs additional computation as we learn the underlying manifold. In order to quantify the effect of the additional computation, we perform wall-clock time analysis and plot average reward vs wall-clock time in Figure 2. Our method outperforms all baselines with similar margins to Figure 1. The trends and shapes of the curves in Figures 1 and 2 are similar. This is not surprising since computation requirements of all the optimizers are rather negligible when compared with the simulation time of MuJoCo.

The only major differences we notice are on the Hopper task. Here the margin between our method and the baselines narrows and the relative ordering of Guided-ES and ARS changes. This is due to the fact that simulation stops when the agent falls. In other words, the simulation time depends on the current solution. Methods that query the simulator for these unstable solutions lose less wall-clock time.

**Quantifying manifold learning performance.** In order to evaluate the learning performance, we project the gradient of the function to the tangent space of the learned manifold and plot the norm of the residual. Since we do not have access to the gradients, we estimate them at 30 time instants, evenly distributed through the learning process. We perform accurate gradient estimation using a very large number of directions (2000). We compute the norm of the residual of the projection as $\frac{1}{T}\sum_{t=1}^{T}\|\nabla_{\mathbf{x}}f(\mathbf{x}^t) - \mathbf{P}_{\mathbf{J}_q(\mathbf{x}^t,\theta^t)}(\nabla_{\mathbf{x}}f(\mathbf{x}^t))\|_2$, where $\mathbf{P}_{\mathbf{A}}(\cdot)$ is projection onto the column space of $\mathbf{A}$. The results are visualized in Figure 3. Our method successfully and quickly learns the manifold in all cases.

| Task | Threshold | LMRS | ARS | CMA-ES | Guided ES | PPO | REMBO | Self Baselines No learning | Offline l. |
|---|---|---|---|---|---|---|---|---|---|
| Swimmer | 325 | **222** | 381 | 460 | 640 | 790 | 520 | 320 | 325 |
| Hopper | 3120 | **2408** | 6108 | 14640 | 5288 | 10397 | (0/5) | 5561 (4/5) | 11616 |
| HalfCheetah | 3430 | **1128** | 4284 | 2888 | 8004 | 6820 | (0/5) | 6528 | 3416 |
| Walker | 4390 | **15525** | 31240 | 96525 | 62544 | 102440 | (0/5) | 295493 (3/5) | 118640 |
| Ant | 3580 | **9240** | 20440 | 154440 | 152400 | 52553 | (0/5) | 24520(4/5) | 63720(2/5) |
| Humanoid | 6000 | **108133** | 220733 | 1923030 | (0/5) | 200430 | (0/5) | 267260(2/5) | 451260(2/5) |

Table 1: Number of episodes required to reach the prescribed threshold on each MuJoCo locomotion task for our method, baselines, and ablations. Lower is better. We average over five random seeds. We denote the number of successful trials as (success/trial) and average over successful trials only. If the number of successful trials is not noted, the method solved the task for all random seeds.

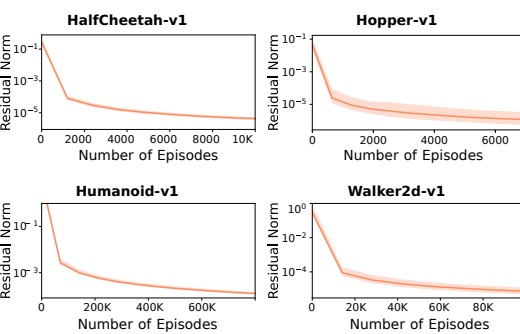

**Ablation studies.** Our method uses three major ideas. i) We learn a manifold that the function lies on. ii) We learn this manifold in an online fashion. iii) We perform random search on the learned manifold. To study the impact of each of these ideas, we perform the following experiments. i) **No learning.** We randomly initialize the manifold $r(\cdot; \theta)$ by sampling the entries of $\theta$ from the standard normal distribution. Then we perform random search on this random manifold. ii) **No online learning.** We collect an offline training dataset by sampling $\mathbf{x}_i$ values uniformly at random from a range that includes the optimal solutions. We evaluate function values at sampled points and learn the manifold. We

Figure 3: Manifold learning accuracy. We plot $\frac{1}{T} \sum_{t=1}^{T} \|\nabla_{\mathbf{x}} f(\mathbf{x}^t) - \mathbf{P}_{\mathbf{J}_q(\mathbf{x}^t, \theta^t)}(\nabla_{\mathbf{x}} f(\mathbf{x}^t))\|_2$. In order to estimate the gradient, we use GRADEST with a high number of directions.

perform random search on this manifold without updating the manifold model. iii) **No search.** We use the gradients of the estimated function ($\nabla_{\mathbf{x}} g(r(\mathbf{x}; \theta^t)\psi^t)$) as surrogate gradients and minimize the function of interest using first-order methods.

We list the results in Table 1. We do not include the no-search baseline since it fails to solve any of the tasks. Failure of the no-search baseline suggests that the estimated functions are powerful enough to guide the search, but not accurate enough for optimization. The no-learning baseline outperforms ARS on the simplest problem (Swimmer), but either fails completely or increases sample complexity on other problems, suggesting that random features are not effective, especially on high-dimensional problems. Although the offline learning baseline solves more tasks than the no-learning one, it has worse sample complexity since initial offline sampling is expensive. This study indicates that all three of the ideas that underpin our method are important.

## 5.2 CONTINUOUS OPTIMIZATION BENCHMARKS

We use continuous optimization problems from the Pagmo problem suite (Biscani et al., 2019). This benchmark includes minimization of 46 functions such as *Rastrigin, Rosenbrock, Schwefel,* etc. (See Appendix B for the complete list.) We use ten random starting points and report the average number of function evaluations required to reach a stationary point. Figure 4 reports the results as performance profiles (Dolan & Moré, 2002). Performance profiles represent how frequently a method is within distance $\tau$ of optimality. Specifically, if we denote the number of function evaluations that method $m$ requires to solve problem $p$ by $T_m(p)$ and the number of function evaluations used by the best method by $T^\star(p) = \min_m T_m(p)$, the performance profile is the fraction of problems for which method $m$ is within distance $\tau$ of the best: $1/N_p \sum_p \mathbb{1}[T_m(p) - T^\star(p) \leq \tau]$, where $\mathbb{1}[\cdot]$ is the indicator function and $N_p$ is the number of problems.

As can be seen in Figure 4, our method outperforms all baselines. The success of our method is not surprising since the functions are typically defined as nonconvex functions of some statistics, inducing manifold structure by construction. REMBO (Bayesian optimization) is close to our method and outperforms the other baselines. We believe this is due to the global geometric structure of the considered functions. Both CMA-ES and Guided-ES outperform ARS.

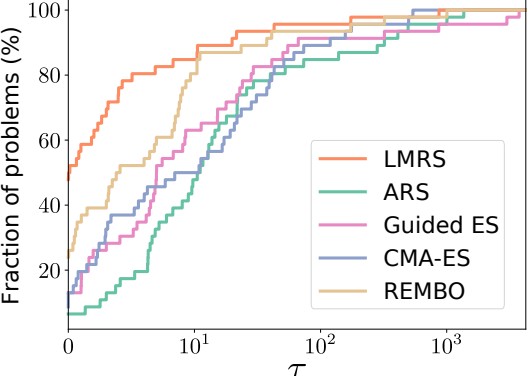

Figure 4: Performance profiles of our method and baselines on an optimization benchmark.

## 5.3 OPTIMIZATION OF AN AIRFOIL

We apply our method to gradient-free optimization of a 2D airfoil. We use a computational fluid dynamics (CFD) simulator, XFoil

| | Init | Airfoil after 1500 Simulations | | | | |
| --- | --- | --- | --- | --- | --- | --- |
| | | LMRS | ARS | Guided ES | CMA-ES | REMBO |
| LIFT | −0.229 | **2.0126** | 0.567 | 1.231 | 1.418 | 1.8645 |
| DRAG | 0.040 | 0.0389 | **0.007** | 0.0598 | 0.0249 | 0.02123 |
| LIFT − DRAG | −0.269 | **1.9737** | 0.560 | 1.1712 | 1.3931 | 1.8432 |
| Foil | | | | | | |

Table 2: Generated airfoils with their lift and drag values after 1500 calls to XFoil (Drela, 1989).

(Drela, 1989), which can simulate an airfoil using its contour plot. We parametrize the airfoils using smooth polynomials of up to 36 degrees. We model the upper and lower parts of the airfoil with different polynomials. The dimensionality of the problem is thus 72. XFoil can simulate various viscosity properties, speeds, and angles of attack. The details are discussed in Appendix B. We plot the resulting airfoil after 1500 simulator calls in Table 2. We also report the lift and drag of the resulting shape. The objective we optimize is LIFT − DRAG. Table 2 suggests that all methods find airfoils that can fly (LIFT > DRAG). Our method yields the highest LIFT − DRAG. Bayesian optimization outperforms the other baselines.

## 5.4 EFFECT OF MANIFOLD AND PROBLEM DIMENSIONALITY

In this section, we perform a controlled experiment to understand the effect of problem dimensionality ($d$) and manifold dimensionality ($n$). We generate a collection of synthetic optimization problems. All synthesized functions follow the manifold hypothesis: $f(\mathbf{x}) = g(r(\mathbf{x}, \theta^\star))$, where $r(\mathbf{x}, \theta^\star)$ is a multilayer perceptron with the architecture Linear$(d, 2n) \to$ ReLU $\to$ Linear$(2n, n)$ and $g(\cdot)$ is a randomly sampled convex quadratic function. In order to sample a convex quadratic function, we sample the parameters of the quadratic function from a Gaussian distribution and project the result to the space of convex quadratic functions.

We choose $d \in \{100, 1000\}$ and plot the objective value with respect to the number of function evaluations for various manifold dimensionalities $n$ in Figure 5. The results suggest that for a given ambient dimensionality $d$, the lower the dimensionality of the data manifold $n$, the more sample-efficient our method. In concordance with our theoretical analysis, the improvement is very significant when $n \ll d$, as can be seen in the cases $n = 5, d = 1000$ and $n = 2, d = 100$.

Interestingly, our method is effective even when the manifold assumption is violated ($n = d$). We hypothesize that this is due to anisotropy in the geometry of the problem. Although all directions are important when $n = d$, some will result in faster search since the function changes more along them. It appears that our method can identify these direction and thus accelerate the search.

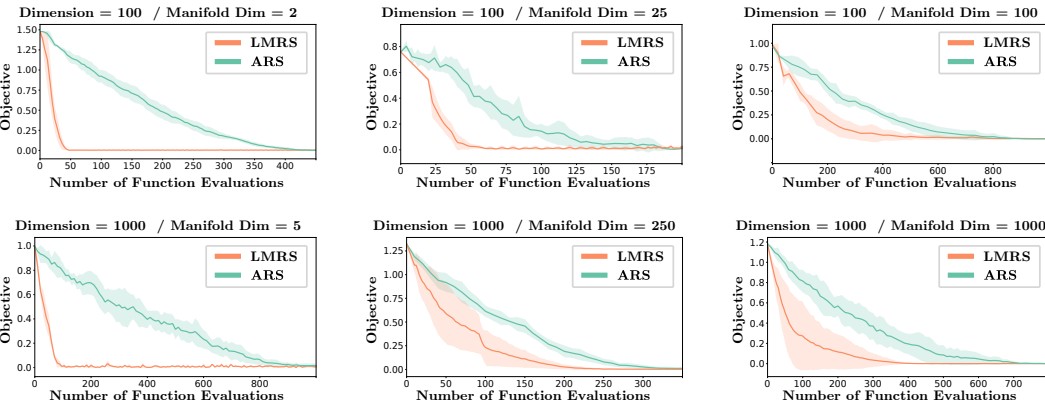

Figure 5: Effect of problem dimensionality and manifold dimensionality. Average objective value over 10 random seeds vs. number of function evaluations. Shaded areas represent 1 standard deviation.

## 6 RELATED WORK

**Derivative-free optimization.** We summarize the work on DFO that is relevant to our paper. For a complete review, readers are referred to Custódio et al. (2017) and Conn et al. (2009). We are specifically interested in random search methods, which have been developed as early as Matyas (1965) and Rechenberg (1973). Convergence properties of these methods have recently been analyzed by Agarwal et al. (2010), Bach & Perchet (2016), Nesterov & Spokoiny (2017), and Dvurechensky et al. (2018). A lower bound on the sample complexity for the convex case has been given by Duchi et al. (2015) and Jamieson et al. (2012). Bandit convex optimization is also highly relevant and we utilize the work of Flaxman et al. (2005) and Shamir (2013).

**Random search for learning continuous control.** Learning continuous control is an active research topic that has received significant interest in the reinforcement learning community. Recently, Salimans et al. (2017) and Mania et al. (2018) have shown that random search methods are competitive with state-of-the-art policy gradient algorithms in this setting. Vemula et al. (2019) analyzed this phenomenon theoretically and characterized the sample complexity of random search and policy gradient methods for continuous control.

**Adaptive random search.** There are various methods in the literature that adapt the search space by using anisotropic covariance as in the case of CMA-ES (Hansen et al., 2003; Hansen, 2016), guided evolutionary search (Maheswaranathan et al., 2019), and active subspace methods (Choromanski et al., 2019). There are also methods that enforce structure such as orthogonality in the search directions (Choromanski et al., 2018). Other methods use information geometry tools as in Wierstra et al. (2014) and Glasmachers et al. (2010). Lehman et al. (2018) use gradient magnitudes to guide neuro-evolutionary search. Staines & Barber (2012) use a variational lower bound to guide the search. In contrast to these methods, we explicitly posit nonlinear manifold structure and directly learn this latent manifold via online learning. Our method is the only one that can learn an arbitrary nonlinear search space given a parametric class that characterizes its geometry.

**Adaptive Bayesian optimization.** Bayesian optimization (BO) is another approach to zeroth-order optimization with desirable theoretical properties (Srinivas et al., 2010). Although we are only interested in methods based on random search, some of the ideas we use have been utilized in BO. Calandra et al. (2016) used the manifold assumption for Gaussian processes. In contrast to our method, they use autoencoders for learning the manifold and assume initial availability of offline data. Similarly, Djolonga et al. (2013) consider the case where the function of interest lies on some linear manifold and collect offline data to identify this manifold. In contrast, we only use online information and our models are nonlinear. Wang et al. (2016) and Kirschner et al. (2019) propose using random low-dimensional features instead of adaptation. Rolland et al. (2018) design adaptive BO methods for additive models. Major distinctions between our work and the adaptive BO literature include our use of nonlinear manifolds, no reliance on offline data collection, and formulation of the problem as online learning.

## 7 CONCLUSION

We presented Learned Manifold Random Search (LMRS): a derivative-free optimization algorithm. Our algorithm learns the underlying geometric structure of the problem online while performing the optimization. Our experiments suggest that LMRS is effective on a wide range of problems and significantly outperforms prior derivative-free optimization algorithms from multiple research communities.

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

## A  PROOFS

### A.1  GRADIENT ESTIMATOR

In this section, we show that when random directions are sampled in the column space of the orthonormal matrix $\mathbf{U}$, perturbations give biased gradient estimates of the manifold smoothed function. Moreover, when $\mathbf{U} = \mathbf{U}^\star$, resulting gradients are unbiased. We formalize this with the following lemma.

**Lemma 1.** *Let $\mathbf{U}^\star$ be an orthonormal basis for the tangent space of the $n$-dimensional manifold $\mathcal{M}$ at point $\mathbf{x} \in \mathcal{M}$, $\mathbf{U}$ be another orthonormal matrix, and $f$ be a function defined on this manifold. Fix $\delta > 0$. Then*

$$\mathbb{E}_{\mathbf{s}\sim\mathbb{S}^{n-1}}[f(\mathbf{x} + \delta\mathbf{U}\mathbf{s})\mathbf{U}\mathbf{s}] = \frac{\delta}{n}\nabla_{\mathbf{x}}\tilde{f}_{\mathbf{U}}(\mathbf{x}) + \mathrm{BIAS}(\mathbf{U})$$

*where $\mathrm{BIAS}(\mathbf{U}) = \mathbb{E}_{\mathbf{s}\in\mathbb{S}^{n-1}}[f(\mathbf{x} + \delta\mathbf{U}\mathbf{s})[\mathbf{U} - \mathbf{U}^\star]\mathbf{s}]$. Moreover, bias is $0$ and the resulting estimator is unbiased when $\mathbf{U} = \mathbf{U}^\star$.*

*Proof.* Without loss of generality, we can assume $\det \mathbf{U} = 1$ and $\det \mathbf{U}^\star = 1$. Using this remark, we can state the proof of the lemma as the straightforward application of the manifold Stoke's theorem

$$\mathbb{E}_{\mathbf{s}\in\mathbb{S}^{n-1}}[f(\mathbf{x} + \delta\mathbf{U}\mathbf{s})\mathbf{U}\mathbf{s}] = \mathbb{E}_{\mathbf{s}\in\mathbb{S}^{n-1}}[f(\mathbf{x} + \delta\mathbf{U}\mathbf{s})\mathbf{U}^\star\mathbf{s}] + \mathbb{E}_{\mathbf{s}\in\mathbb{S}^{n-1}}[f(\mathbf{x} + \delta\mathbf{U}\mathbf{s})[\mathbf{U} - \mathbf{U}^\star]\mathbf{s}]$$

$$\overset{(a)}{=} \frac{1}{\mathrm{vol}(\delta\mathbb{S}^{n-1})}\int_{\delta\mathbb{S}^{n-1}} f(\mathbf{x} + \delta\mathbf{U}\mathbf{s})\mathbf{U}^\star\mathbf{s}\,d\mathbf{s} + \underbrace{\mathbb{E}_{\mathbf{s}\in\mathbb{S}^{n-1}}[f(\mathbf{x} + \delta\mathbf{U}\mathbf{s})[\mathbf{U} - \mathbf{U}^\star]\mathbf{s}]}_{\mathrm{BIAS}(\mathbf{U})}$$

$$= \frac{1}{\mathrm{vol}(\delta\mathbb{S}^{n-1})}\int_{\delta\mathbb{S}^{n-1}} f(\mathbf{x} + \delta\mathbf{U}\mathbf{s})\mathbf{U}^\star \det\mathbf{U}^\star\frac{\mathbf{s}}{\|\mathbf{s}\|}\,d\mathbf{s} + \mathrm{BIAS}(\mathbf{U})$$

$$\overset{(b)}{=} \frac{1}{\mathrm{vol}(\delta\mathbb{S}^{n-1})}\int_{\delta\mathbb{S}^{n-1}} f(\mathbf{x} + \delta\mathbf{U}\mathbf{U}^{\star\mathsf{T}}\mathbf{U}^\star\mathbf{s})\mathbf{U}^\star \det\mathbf{U}^\star\frac{\mathbf{s}}{\|\mathbf{s}\|}\,d\mathbf{s} + \mathrm{BIAS}(\mathbf{U}) \qquad (8)$$

$$\overset{(c)}{=} \frac{1}{\mathrm{vol}(\delta\mathbb{S}^{n-1})}\nabla_{\mathbf{x}}\int_{\delta\mathbb{B}^n} f(\mathbf{x} + \delta\mathbf{U}\mathbf{U}^{\star\mathsf{T}}\mathbf{U}^\star\mathbf{v})\,d\mathbf{v} + \mathrm{BIAS}(\mathbf{U})$$

$$\overset{(d)}{=} \frac{\mathrm{vol}(\delta\mathbb{B}^n)}{\mathrm{vol}(\delta\mathbb{S}^{n-1})}\nabla_{\mathbf{x}}\frac{\int_{\delta\mathbb{B}^n} f(\mathbf{x} + \delta\mathbf{U}\mathbf{v})\,d\mathbf{v}}{\mathrm{vol}(\delta\mathbb{B}^n)} + \mathrm{BIAS}(\mathbf{U})$$

$$\overset{(e)}{=} \frac{\delta}{n}\nabla_{\mathbf{x}}\tilde{f}_{\mathbf{U}}(\mathbf{x}) + \mathrm{BIAS}(\mathbf{U}).$$

where vol denotes volume, and we use the definition of the expectation in (a, e), orthonormality of $\mathbf{U}^\star$ in (b,d), manifold Stoke's theorem in (c) and the fact that the ratio of volume to the surface area of a $n-$dimensional ball of radius $\delta$ is $\frac{\delta}{n}$ in (e). Moreover, bias vanishes when $\mathbf{U} = \mathbf{U}^\star$.  □

### A.2  SAMPLE COMPLEXITY FOR RANDOM SEARCH AND MANIFOLD RANDOM SEARCH

In this section, we bound the sample complexity of the random search (Algorithm 1) and the manifold random search (Algorithm 2). Our analysis starts with studying the relationship between the function ($f$) and its smoothed ($\hat{f}$) as well as manifold smoothed ($\tilde{f}_{\mathbf{U}}$) versions in section A.2.2. We show that $L-$Lipschitzness and $\mu-$smoothness of the function extend to the smoothed functions. Moreover, we also bound the difference between the gradients of the original function and the gradients of the smoothed versions. Next, we study the second moment of the gradient estimator in section A.2.3. Finally, we state the sample complexity of SGD on non-convex functions in section A.2.1. Combining these results, we state the final sample complexity of random search and manifold random search in section A.2.4&A.2.5.

### A.2.1  CONVERGENCE OF SGD FOR NON-CONVEX FUCTIONS

The convergence of the SGD has been widely studied and here we state its convergence result for non-convex functions from Ghadimi & Lan (2013) as a Lemma and give its proof for the sake of completeness.

**Lemma 2** (Convergence of SGD (Ghadimi & Lan, 2013; Vemula et al., 2019)). *Consider running SGD on $f(\mathbf{x})$ that is $\mu$-smooth and $L$-Lipschitz for $T$ steps starting with initial solution $\mathbf{x}^0$. Denote $\Omega^0 = f(\mathbf{x}^0) - f(\mathbf{x}^\star)$ where $\mathbf{x}^\star$ is the globally optimal point and assume that the unbiased gradient estimate has second moment bounded with $V$. Then,*

$$\frac{1}{T} \sum_{t=1}^{T} \mathbb{E}\|\nabla_\mathbf{x} f(\mathbf{x}^t)\|_2^2 \leq \sqrt{\frac{8\Omega^0 \mu V}{T}} \tag{9}$$

*Proof.* We denote the step size as $\alpha$ and the unbiased gradient estimate as $\mathbf{g}^t$. We analyze the step at $t$ as;

$$f(\mathbf{x}^{t+1}) = f(\mathbf{x}^t - \alpha\mathbf{g}^t)$$
$$\leq f(\mathbf{x}^t) - \alpha\nabla_\mathbf{x} f(\mathbf{x}^t)^\intercal \mathbf{g}^t + \frac{\mu\alpha^2}{2}\|\mathbf{g}^t\|_2^2 \tag{10}$$

where we used the $\mu-$smoothness of the function. Taking expectation of the inequality,

$$\mathbb{E}_{\mathbf{g}^t}[f(\mathbf{x}^{t+1})] \leq f(\mathbf{x}^t) - \alpha\|\nabla_\mathbf{x} f(\mathbf{x}^t)\|_2^2 + \frac{\mu\alpha^2}{2}\mathbb{E}[\|\mathbf{g}^t\|_2^2] \tag{11}$$

Using the bounded second moment of the gradient, and summing from step 1 to $T$,

$$\sum_{t=1}^{T} \mathbb{E}_{\mathbf{g}^t}[f(\mathbf{x}^{t+1})] \leq \sum_{t=1}^{T} f(\mathbf{x}^t) - \alpha\sum_{t=1}^{T}\|\nabla_\mathbf{x} f(\mathbf{x}^t)\|_2^2 + \frac{\mu\alpha^2 TV}{2} \tag{12}$$

Re-arranging the terms, we obtain,

$$\sum_{t=1}^{T}\|\nabla_\mathbf{x} f(\mathbf{x}^t)\|_2^2 \leq \frac{1}{\alpha}\mathbb{E}_{\mathbf{g}^0,\ldots,\mathbf{g}^t}[f(\mathbf{x}^0) - f(\mathbf{x}^{t+1})] + \frac{\mu\alpha TV}{2}$$
$$\leq \frac{\Delta^0}{\alpha} + \mu T\alpha V \tag{13}$$

Set $\alpha = \sqrt{\frac{2\Delta^0}{\mu TV}}$, and divide the inequality to $T$ in order to obtain the required inequality as

$$\frac{1}{T}\sum_{t=1}^{T}\|\nabla_\mathbf{x} f(\mathbf{x}^t)\|_2^2 \leq \sqrt{\frac{8\Delta^0 \mu V}{T}}. \tag{14}$$

$\square$

### A.2.2 PRELIMINARY RESULTS ON SMOOTHED FUNCTIONS

First, we will show that the $\mu-$smoothness and $L-$Lipschitness properties of $f$ applies to $\hat{f}$ and $\tilde{f}$.

$$|\tilde{f}_\mathbf{U}(\mathbf{x}_1) - \tilde{f}_\mathbf{U}(\mathbf{x}_2)| = |E_{\mathbf{v}\in\mathbb{B}^n}[f(\mathbf{x}_1 + \delta\mathbf{U}\mathbf{v})] - E_{\mathbf{v}\in\mathbb{B}^n}[f(\mathbf{x}_2 + \delta\mathbf{U}\mathbf{v})]|$$
$$= |E_{\mathbf{v}\in\mathbb{B}^n}[f(\mathbf{x}_1 + \delta\mathbf{U}\mathbf{v}) - f(\mathbf{x}_2 + \delta\mathbf{U}\mathbf{v})]|$$
$$\leq |E_{\mathbf{v}\in\mathbb{B}_n}[L\|\mathbf{x}_1 - \mathbf{x}_2\|_2]| \tag{15}$$
$$\overset{(a)}{=} L\|\mathbf{x}_1 - \mathbf{x}_2\|_2$$

where we use $L-$Lipschitz continuity of $f$ in (a), and,

$$\|\nabla\tilde{f}_\mathbf{U}(\mathbf{x}_1) - \nabla\tilde{f}_\mathbf{U}(\mathbf{x}_2)\|_2 = \|\nabla E_{\mathbf{v}\in\mathbb{B}^n}[f(\mathbf{x}_1 + \delta\mathbf{U}\mathbf{v})] - \nabla E_{\mathbf{v}\in\mathbb{B}^n}[f(\mathbf{x}_2 + \delta\mathbf{U}\mathbf{v})]\|_2$$
$$= \|E_{\mathbf{v}\in\mathbb{B}^n}[\nabla f(\mathbf{x}_1 + \delta\mathbf{U}\mathbf{v})] - E_{\mathbf{v}\in\mathbb{B}^n}[\nabla f(\mathbf{x}_2 + \delta\mathbf{U}\mathbf{v})]\|_2$$
$$= \|E_{\mathbf{v}\in\mathbb{B}^n}[\nabla f(\mathbf{x}_1 + \delta\mathbf{U}\mathbf{v}) - \nabla f(\mathbf{x}_2 + \delta\mathbf{U}\mathbf{v})]\|_2 \tag{16}$$
$$\overset{(b)}{\leq} E_{\mathbf{v}\in\mathbb{B}^n}[\|\nabla f(\mathbf{x}_1 + \delta\mathbf{U}\mathbf{v}) - \nabla f(\mathbf{x}_2 + \delta\mathbf{U}\mathbf{v})\|_2]$$
$$\overset{(c)}{\leq} \mu\|\mathbf{x}_1 - \mathbf{x}_2\|_2$$

where we use Jensen's inequality and convexity of the norm in (b) and $\mu-$smoothness of $f$ in (c). Hence, $\mu-$smoothness and $L-$ Lipschitness applies to $\tilde{f}_{\mathbf{U}}$ for any $\mathbf{U}$. Take $n = d$ and $\mathbf{U} = \mathbf{I}$, then the $\mu-$smoothness and $L-$ Lipschitness applies to $\hat{f}$.

Next, we will study the impact of using the gradients of the smoothed function instead of the original function.

$$
\begin{aligned}
\frac{1}{T}\sum_{t=1}^{T}\|\nabla_{\mathbf{x}}f(\mathbf{x})\|_2^2 &= \frac{1}{T}\sum_{t=1}^{T}\|\nabla_x f(\mathbf{x}) - \nabla_x \tilde{f}_{\mathbf{U}}(\mathbf{x}) + \nabla_{\mathbf{x}}\tilde{f}_{\mathbf{U}}(\mathbf{x})\|_2^2 \\
&\leq \frac{2}{T}\sum_{t=1}^{T}\|\nabla_x f(\mathbf{x}) - \nabla_{\mathbf{x}}\tilde{f}_{\mathbf{U}}(\mathbf{x})\|_2^2 + \frac{2}{T}\sum_{t=1}^{T}\|\nabla_{\mathbf{x}}\tilde{f}_{\mathbf{U}}(\mathbf{x})\|_2^2.
\end{aligned}
\tag{17}
$$

where we use $\|a + b\|_2^2 \leq 2\|a\|_2^2 + 2\|b\|_2^2$. We further bound the left term as

$$
\begin{aligned}
\|\nabla_x f(\mathbf{x}) - \nabla_{\mathbf{x}}\tilde{f}_{\mathbf{U}}(\mathbf{x})\|_2^2 &= \|\nabla_x f(\mathbf{x}) - \nabla_{\mathbf{x}}\mathbb{E}_{\mathbf{v}\sim\mathbb{B}_n}[f(\mathbf{x} + \delta\mathbf{U}\mathbf{v})]\|_2^2 \\
&\overset{(a)}{=} \|\mathbb{E}_{\mathbf{v}\sim\mathbb{B}_n}[\nabla_{\mathbf{x}}f(\mathbf{x}) - \nabla_{\mathbf{x}}f(\mathbf{x} + \delta\mathbf{U}\mathbf{v})]\|_2^2 \\
&\overset{(b)}{\leq} \|\mathbb{E}_{\mathbf{v}\sim\mathbb{B}_n}[\delta\mu\|\mathbf{U}\mathbf{v}\|_2]\|_2^2 \\
&\overset{(c)}{\leq} \delta^2\mu^2
\end{aligned}
\tag{18}
$$

using dominated convergence theorem in (a), the $\mu-$smoothness of $f$ in (b) and orthonormality of $\mathbf{U}$ and the fact that norm of any point in a unit ball is bounded by 1. By taking $n = d$ and $\mathbf{U} = \mathbf{I}$, this result also implies the same for $\hat{f}$. Hence,

$$
\begin{aligned}
\frac{1}{T}\sum_{t=1}^{T}\|\nabla_{\mathbf{x}}f(\mathbf{x})\|_2^2 &\leq \frac{2}{T}\sum_{t=1}^{T}\|\nabla_{\mathbf{x}}\hat{f}(\mathbf{x})\|_2^2 + 2\delta^2\mu^2 \\
\frac{1}{T}\sum_{t=1}^{T}\|\nabla_{\mathbf{x}}f(\mathbf{x})\|_2^2 &\leq \frac{2}{T}\sum_{t=1}^{T}\|\nabla_{\mathbf{x}}\tilde{f}_{\mathbf{U}^t}(\mathbf{x})\|_2^2 + 2\delta^2\mu^2 \quad \forall_{\mathbf{U}^1,...,\mathbf{U}^T}
\end{aligned}
\tag{19}
$$

### A.2.3 SECOND MOMENT OF THE GRADIENT ESTIMATOR

We will start with studying the second moment of our gradient estimate for the manifold case. We bound the expected square norm of the gradient estimate as

$$
\begin{aligned}
\mathbb{E}_{\mathbf{s}\in\mathbb{S}^n,\xi}\left[\|\mathbf{g}_m^t\|_2^2\right] &= \mathbb{E}_{\mathbf{s}\in\mathbb{S}^n,\xi}\left[\left\|\frac{n}{2\delta}\left[F(\mathbf{x}^t + \delta\mathbf{U}\mathbf{s}, \xi_1) - F(\mathbf{x}^t - \delta\mathbf{U}\mathbf{s}, \xi_2)\right]\mathbf{U}\mathbf{s}\right\|_2^2\right] \\
&\overset{(a)}{=} \frac{n^2}{4\delta^2}\mathbb{E}_{\mathbf{s}\in\mathbb{S}^n,\xi}\left[\left(F(\mathbf{x}^t + \delta\mathbf{U}\mathbf{s}, \xi_1) - F(\mathbf{x} - \delta\mathbf{U}\mathbf{s}, \xi_2)\right)^2\right] \\
&\overset{(b)}{\leq} \frac{n^2}{2\delta^2}\mathbb{E}_{\mathbf{s}\in\mathbb{S}^n}\left[(f(\mathbf{x} + \delta\mathbf{U}\mathbf{s}) - f(\mathbf{x} - \delta\mathbf{U}\mathbf{s}))^2\right] \\
&\quad + \frac{n^2}{\delta^2}\mathbb{E}_{\mathbf{s}\in\mathbb{S}^n,\xi}\left[(F(\mathbf{x} + \delta\mathbf{U}\mathbf{s}, \xi) - f(\mathbf{x} + \delta\mathbf{U}\mathbf{s}))^2\right] \\
&\quad + \frac{n^2}{\delta^2}\mathbb{E}_{\mathbf{s}\in\mathbb{S}^n,\xi}\left[(F(\mathbf{x} - \delta\mathbf{U}\mathbf{s}, \xi) - f(\mathbf{x} - \delta\mathbf{U}\mathbf{s}))^2\right] \\
&\overset{(c)}{\leq} 2n^2L^2 + \frac{2n^2V_F}{\delta^2}
\end{aligned}
\tag{20}
$$

where we use orthonormality of $\mathbf{U}$ and unit norm property of $\mathbf{s}$ in (a), add and substract $f(\mathbf{x} + \delta\mathbf{U}\mathbf{s}) - f(\mathbf{x} - \delta\mathbf{U}\mathbf{s})$ and use $(a + b)^2 \leq 2a^2 + 2b^2$ in (b), use the bounded variance of $F$ and the Lipschitz smoothness of $f$ in (c).

Second moment of the random search estimator can also computed similarly. And, the resulting bound would be

$$\mathbb{E}_{\mathbf{s} \in \mathbb{S}^n, \xi} \left[ \|\mathbf{g}_e^t\|_2^2 \right] \leq 2d^2 L^2 + \frac{2d^2 V_F}{\delta^2} \tag{21}$$

### A.2.4 PROOF OF PROPOSITION 1

*Proof.* Analysis of SGD from Lemma 2 shows that

$$\frac{1}{T} \sum_{t=1}^T \|\nabla_{\mathbf{x}} f(\mathbf{x}^t)\|_2^2 \leq \sqrt{\frac{8\Omega^0 \mu V}{T}}. \tag{22}$$

Using the bound on the second moment of the estimator we derive in (21),

$$\frac{1}{T} \sum_{t=1}^T \|\nabla_{\mathbf{x}} f(\mathbf{x}^t)\|_2^2 \leq \sqrt{\frac{8\Omega^0 \mu}{T}} \sqrt{2d^2 L^2 + \frac{2d^2 V_F}{\delta^2}} \tag{23}$$

Using the relationship between $f$ and $\hat{f}$ we derive in (19),

$$\frac{1}{T} \sum_{t=1}^T \|\nabla_{\mathbf{x}} \hat{f}(\mathbf{x})\|_2^2 \leq 2\delta^2 \mu^2 + 2\sqrt{\frac{8\Omega^0 \mu}{T}} \sqrt{2d^2 L^2 + \frac{2d^2 V_F}{\delta^2}}. \tag{24}$$

We use the property $\sqrt{a+b} \leq \sqrt{a} + \sqrt{b}$, and solve for $\alpha$. After we substituted the resulting $\delta$,

$$\frac{1}{T} \sum_t \|\nabla_{\mathbf{x}} f(\mathbf{x}^t)\|_2^2 \leq \frac{c_0 + c_1 d}{T^{\frac{1}{2}}} + \frac{c_2 d^{\frac{2}{3}}}{T^{\frac{1}{3}}} \tag{25}$$

where $c_0 = 4L\sqrt{2\Omega^0 \mu}$, $c_1 = \sqrt{2} c_0$, and $c_2 = (4 V_F \Omega^0)^{\frac{1}{3}} (2\mu + 4\mu^{\frac{5}{6}})$. $\qquad\square$

### A.2.5 PROOF OF PROPOSITION 2

*Proof.* Sample complexity of the manifold random search follows closely the proof of Proposition 1. We summarize here for the sake of completeness. Using the analysis of SGD from Lemma 2,

$$\frac{1}{T} \sum_{t=1}^T \|\nabla_{\mathbf{x}} f(\mathbf{x}^t)\|_2^2 \leq \sqrt{\frac{8\Omega^0 \mu V}{T}}. \tag{26}$$

Using the bound on the second moment of the estimator we derive in (21), and the relationship between $f$ and $\tilde{f}_{\mathbf{U}^\star}$ we derive in (19),

$$\frac{1}{T} \sum_{t=1}^T \|\nabla_{\mathbf{x}} \hat{f}(\mathbf{x})\|_2^2 \leq 2\delta^2 \mu^2 + 2\sqrt{\frac{8\Omega^0 \mu}{T}} \sqrt{2n^2 L^2 + \frac{2n^2 V_F}{\delta^2}}. \tag{27}$$

We first use the property $\sqrt{a+b} \leq \sqrt{a} + \sqrt{b}$, then solve for $\alpha$ and $\delta$. Finally, we substitute resulting $\delta$ to get the final result as;

$$\frac{1}{T} \sum_t \|\nabla_{\mathbf{x}} f(\mathbf{x}^t)\|_2^2 \leq \frac{c_0 + c_1 n}{T^{\frac{1}{2}}} + \frac{c_2 n^{\frac{2}{3}}}{T^{\frac{1}{3}}} \tag{28}$$

where $c_0 = 4L\sqrt{2\Omega^0 \mu}$, $c_1 = \sqrt{2} c_0$, and $c_2 = (4 V_F \Omega^0)^{\frac{1}{3}} (2\mu + 4\mu^{\frac{5}{6}})$.

$\qquad\square$

### A.3 PROOF OF THE THEOREM 1

*Proof.* We will prove our main theorem using a three major arguments. First, we analyze the SGD of a non-convex function with biased gradients in section A.3.1. Second, we show that the expected value of our loss function is equal to the bias term in section A.3.2. In order to bound the difference between the empirical loss function we minimize and its expectation, we use the Freedman's inequality Freedman (1975). Third, we bound the empirical loss in section A.3.3 in terms of the distance travelled by the iterates of the optimization $\|x^{t+1} - x^t\|_2$. Finally, we optimize the resulting bound in terms of the finite difference step ($\delta^t$), SGD step size ($\alpha^t$), and mixing coefficients ($\beta$) to obtain the final statement of the theorem in section A.3.4.

### A.3.1 ANALYSIS OF SGD WITH BIAS

In order to analyze the SGD with bias, we will denote the gradient at the iteration $t$, as $\mathbf{g}^t$. Moreover, we will assume that its bias is $\mathbf{b}^t$ as $\mathbb{E}[\mathbf{g}^t] = \mathbf{b}^t + \nabla_{\mathbf{x}}\tilde{F}(\mathbf{x}, \xi)$. Using the $\mu-$smoothness of the function $\tilde{F}$, we can state that

$$\tilde{F}(\mathbf{x}^{t+1}, \xi) = \tilde{F}(\xi^t - \alpha\mathbf{g}^t, \xi) \leq \tilde{F}(\mathbf{x}^t, \xi) - \alpha\nabla_{\mathbf{x}}\tilde{F}(\mathbf{x}^t, \xi)\mathbf{g}^t + \frac{\mu\alpha^2}{2}\|\mathbf{g}^t\|_2^2. \quad (29)$$

Let's assume the effect of the bias is bounded as $|\nabla_{\mathbf{x}}\tilde{F}(\mathbf{x}^t, \xi)^\mathsf{T}\mathbf{b}^t| \leq B^t$, then

$$\tilde{F}(\mathbf{x}^{t+1}, \xi) \leq \tilde{F}(\mathbf{x}^t, \xi) - \alpha\nabla_{\mathbf{x}}\tilde{F}(\mathbf{x}^t, \xi)[\mathbf{g}^t - \mathbf{b}^t] + \frac{\mu\alpha^2}{2}\|\mathbf{g}^t\|_2^2 + \alpha B^t. \quad (30)$$

Taking the expectation with respect to $\mathbf{s}$ and $\xi$, we get

$$\alpha\|\nabla_{\mathbf{x}}\tilde{f}(\mathbf{x}^t)\|_2^2 \leq \mathbb{E}_{\mathbf{s},\xi}[\tilde{f}(\mathbf{x}^{t+1})] - \tilde{f}(\mathbf{x}^t) + \frac{\mu\alpha^2\mathbf{V_g}}{2} + \alpha B^t \quad (31)$$

where $\mathbf{V_g} = \mathbb{E}[\|\mathbf{g}^t\|_2^2]$. Summing up from $t = 1$ to $T$ and dividing by $\alpha$, we obtain

$$\sum_{t=1}^{T}\|\nabla_{\mathbf{x}}\tilde{f}(\mathbf{x}^t)\|_2^2 \leq \frac{\Omega}{\alpha} + \frac{\mu\alpha T\mathbf{V_g}}{2} + \sum_{t=1}^{T}B^t. \quad (32)$$

We now compute the bound on the bias term ($B^t$) using Lemma 1 and $\mathbf{g}^t = \beta\mathbf{g}_e^t + (1 - \beta)\mathbf{g}_m^t$,

$$\nabla_{\mathbf{x}}\tilde{F}(\mathbf{x}^t, \xi)^\mathsf{T}\mathbf{b}^t = (1 - \beta)\mathbb{E}_{\mathbf{s}\in\mathbb{S}^{n-1}}\left[F(\mathbf{x} + \delta\mathbf{Us}, \xi)\nabla_{\mathbf{x}}\tilde{F}(\mathbf{x}^t, \xi)^\mathsf{T}[\mathbf{U} - \mathbf{U}^\star]\mathbf{s}\right]$$
$$\leq (1 - \beta)\Omega\mathbb{E}_{\mathbf{s}\in\mathbb{S}^{n-1}}\left[|\nabla_{\mathbf{x}}\tilde{F}(\mathbf{x}^t, \xi)^\mathsf{T}[\mathbf{U} - \mathbf{U}^\star]\mathbf{s}|\right] \quad (33)$$

where we used the fact that function is bounded as $F(\mathbf{x}, \xi) \leq \Omega$. Since $\nabla_{\mathbf{x}}\tilde{F}(\mathbf{x}, \xi)$ lies in the column space of $\mathbf{U}^\star$, for some $\mathbf{p}$,

$$\mathbf{p}^\mathsf{T}\mathbf{U}^{\star\mathsf{T}}[\mathbf{U} - \mathbf{U}^\star] = \mathbf{p}^\mathsf{T}[\mathbf{U}^{\star\mathsf{T}}\mathbf{U} - \mathbf{I}] = \mathbf{p}^\mathsf{T}[\mathbf{U}^{\star\mathsf{T}}\mathbf{U} - \mathbf{U}^\mathsf{T}\mathbf{U}] = [\nabla_{\mathbf{x}}\tilde{F}(\mathbf{x}, \xi)^\mathsf{T} - \mathbf{p}^\mathsf{T}\mathbf{U}^\mathsf{T}]\mathbf{U} \quad (34)$$

Choice of bases ($\mathbf{U}^\star$) is not unique in Lemma 1. It can be any orthonormal basis spanning the tangent space of the manifold. In order to minimize $\mathbb{E}_{\mathbf{s}\in\mathbb{S}^{n-1}}\left[|\nabla_{\mathbf{x}}\tilde{F}(\mathbf{x}^t, \xi)^\mathsf{T}[\mathbf{U} - \mathbf{U}^\star]\mathbf{s}|\right]$, choose the basis which will set $\mathbf{Up}$ to the projection of $\nabla_{\mathbf{x}}\tilde{F}(\mathbf{x}^t, \xi)$ into the column space of $\mathbf{U}$. Then,

$$\|\nabla_{\mathbf{x}}\tilde{F}(\mathbf{x}^t, \xi)^\mathsf{T}[\mathbf{U} - \mathbf{U}^\star]\|_2 \leq \min_{\mathbf{q}\in\mathbb{R}^n}\|\nabla_{\mathbf{x}}\tilde{F}(\mathbf{x}^t, \xi) - \mathbf{Uq}\|_2 \leq \|\nabla_{\mathbf{x}}\tilde{F}(\mathbf{x}^t, \xi) - \nabla_{\mathbf{x}}g(r(\mathbf{x}^t; \theta^t); \psi^t)\|_2 \quad (35)$$

where last inequality is due to $\nabla_{\mathbf{x}}g(r(\mathbf{x}^t; \theta^t); \psi^t)$ being in the column space of $\mathbf{U}$. Combining with (32), (19), and $0 \leq \beta \leq 1$;

$$\frac{1}{T}\sum_{t=1}^{T}\|\nabla_{\mathbf{x}}f(\mathbf{x}^t)\|_2^2 \leq \frac{\Omega}{\alpha T} + \frac{\mu\alpha\mathbf{V_g}}{2} + \frac{\Omega}{T}\sum_{t=1}^{T}\|\nabla_{\mathbf{x}}\tilde{F}(\mathbf{x}^t, \xi) - \nabla_{\mathbf{x}}g(r(\mathbf{x}^t; \theta^t); \psi^t)\|_2 + \delta^2\mu^2. \quad (36)$$

We proceed to bound $\sum_{t=1}^{T}\|\nabla_{\mathbf{x}}\tilde{F}(\mathbf{x}^t, \xi) - \nabla_{\mathbf{x}}g(r(\mathbf{x}^t; \theta^t); \psi^t)\|_2$ in the next section.

### A.3.2 ROLE OF THE BANDIT FEEDBACK

The true loss we are interested in is the effect of bias on the SGD. The bias is the sum of the differences between the gradients of the true function ($\tilde{F}(\mathbf{x}, \xi)$) and the estimated one ($g(r(\mathbf{x}; \theta); \psi)$) as derived in (36). On the other hand, the empirical information we have is the projection of this loss to a random direction ($\mathbf{s}$) with an additional noise term. In this section, we will analyze the difference between the bias and the empirical loss without the noise. We will include the discussion on the noise in section A.3.3.

First, we will show that expectation of the empirical loss over a direction uniformly chosen from a unit sphere is the bias term. In order to show this, we need an elementary result which is $\mathbb{E}_{\mathbf{s} \in \mathbb{S}^{d-1}}[(\mathbf{s}^t \mathbf{v})^2] = \frac{\|\mathbf{v}\|^2}{d}$. We show this result in Section A.4.1. Using this result,

$$\mathbb{E}_{\mathbf{s} \in \mathbb{S}^{d-1}}\left[\left(\mathbf{s}^{\mathsf{T}}\left(\nabla_{\mathbf{x}}\tilde{F}(\mathbf{x},\xi) - \nabla_{\mathbf{x}}g((r(\mathbf{x};\theta);\psi))\right)\right)^2\right] = \frac{1}{d}\left\|\nabla_{\mathbf{x}}\tilde{F}(\mathbf{x},\xi) - \nabla_{\mathbf{x}}g((r(\mathbf{x};\theta);\psi))\right\|_2^2 \quad (37)$$

We introduce the notation $\Delta(\mathbf{s},\mathbf{x},\xi,\theta,\psi) = \left(\mathbf{s}^{\mathsf{T}}\left(\nabla_{\mathbf{x}}\tilde{F}(\mathbf{x},\xi) - \nabla_{\mathbf{x}}g((r(\mathbf{x};\theta);\psi))\right)\right)^2$ for clarity and, proceed to bound the difference $\left|\Delta(\mathbf{s},\mathbf{x},\xi,\theta,\psi) - \mathbb{E}_{\mathbf{s}}[\Delta(\mathbf{s},\mathbf{x},\xi,\theta,\psi)]\right|$. Consider the sequence of differences as,

$$\mathbf{Z}^t = \Delta(\mathbf{s}^t,\mathbf{x}^t,\xi^t,\theta^t,\psi^t) - \frac{1}{d}\left\|\nabla_{\mathbf{x}}\tilde{F}(\mathbf{x}^t,\xi^t) - \nabla_{\mathbf{x}}g((r(\mathbf{x}^t;\theta^t);\psi^t))\right\|_2^2, \quad (38)$$

it is clear that $\mathbb{E}[\mathbf{Z}^t] = 0$ for all $t$. Moreover, the differences are bounded due to the Lipschitz continuity. Hence, $\mathbf{Z}^t$ is a martingale difference sequence. We use the Freedman's inequality (Freedman, 1975) in order to bound $\sum \mathbf{Z}^t$, similar to the seminal work studying the generalization of online learning by Kakade & Tewari (2009). Freedman's inequality (Freedman, 1975) states that if $\mathbf{Z}^t$ is a martingale difference sequence,

$$P\left(\sum_{t=1}^{T}\mathbf{Z}^t \geq \max\left\{2\sqrt{\sum_{i=1}^{T}Var(\mathbf{Z}^t)}, 3b\sqrt{\ln(1/\gamma)}\right\}\sqrt{\ln(1/\gamma)}\right) \leq 4\gamma\ln(T) \quad (39)$$

where $b$ is the bound on $\mathbf{Z}^t$ as $|\mathbf{Z}^t| \leq b$ for all $t$. Before we substitute the $\mathbf{Z}^t$ in the Freedman's inequality, we need to compute the variance of the $\mathbf{Z}^t$. We bound the variance using the definition of the variance as

$$\mathbb{E}_{\mathbf{s} \in \mathbb{S}^{d-1}}\left[\left(\left(\mathbf{s}^{t\mathsf{T}}\left(\nabla_{\mathbf{x}}\tilde{F}(\mathbf{x}^t,\xi^t) - \nabla_{\mathbf{x}}g((r(\mathbf{x}^t;\theta^t);\psi^t))\right)\right)^2 - \frac{1}{d}\left\|\nabla_{\mathbf{x}}\tilde{F}(\mathbf{x}^t,\xi^t) - \nabla_{\mathbf{x}}g((r(\mathbf{x}^t;\theta^t);\psi^t)\|^2\right)^2\right]$$

$$\leq \mathbb{E}_{\mathbf{s} \in \mathbb{S}^{d-1}}\left[\left(\frac{1}{d}\left(\nabla_{\mathbf{x}}\tilde{F}(\mathbf{x}^t,\xi^t) - \nabla_{\mathbf{x}}g((r(\mathbf{x}^t;\theta^t);\psi^t))\right)^{\mathsf{T}}\left(d\mathbf{s}^t - \nabla_{\mathbf{x}}\tilde{F}(\mathbf{x}^t,\xi^t) + \nabla_{\mathbf{x}}g((r(\mathbf{x}^t;\theta^t);\psi^t))\right)\right)^2\right]$$

$$\overset{(a)}{\leq} \frac{\left\|\nabla_{\mathbf{x}}\tilde{F}(\mathbf{x}^t,\xi^t) - \nabla_{\mathbf{x}}g((r(\mathbf{x}^t;\theta^t);\psi^t))\right\|_2^2}{d^2}\mathbb{E}_{\mathbf{s} \in \mathbb{S}^{d-1}}\left[\left(\nu^{\mathsf{T}}\left(d\mathbf{s}^t - \nabla_{\mathbf{x}}\tilde{F}(\mathbf{x}^t,\xi^t) + \nabla_{\mathbf{x}}g((r(\mathbf{x}^t;\theta^t);\psi^t))\right)\right)^2\right]$$

$$\overset{(b)}{\leq} \frac{1}{d^2}\left\|\nabla_{\mathbf{x}}\tilde{F}(\mathbf{x}^t,\xi^t) - \nabla_{\mathbf{x}}g((r(\mathbf{x}^t;\theta^t);\psi^t))\right\|_2^2\left(d^2\mathbb{E}_{\mathbf{s} \in \mathbb{S}^{d-1}}[(\nu^{\mathsf{T}}\mathbf{s}^t)^2] + 2L^2\right)$$

$$= \left(\frac{2L^2 + d}{d^2}\right)\left\|\nabla_{\mathbf{x}}\tilde{F}(\mathbf{x}^t,\xi^t) - \nabla_{\mathbf{x}}g((r(\mathbf{x}^t;\theta^t);\psi^t))\right\|_2^2$$

$$(40)$$

where we denote the unit vector in the direction of $\nabla_{\mathbf{x}}\tilde{F}(\mathbf{x}^t,\xi^t) - \nabla_{\mathbf{x}}g((r(\mathbf{x}^t;\theta^t);\psi^t))$ as $\nu$ in (a) and use the Lipschitz property as well as (63) in (b). Finally we substitute this result in Freedman's inequality and use the expectation $\Delta(\mathbf{s}^t,\mathbf{x}^t,\xi^t,\theta^t,\psi^t)$ from (37). We also introduce the shorthand notation $\Delta^t = \Delta(\mathbf{s}^t,\mathbf{x}^t,\xi^t,\theta^t,\psi^)$. With probability at least $1 - 4\gamma\ln(T)$,

$$\sum_{t=1}^{T}\mathbb{E}[\Delta^t] \leq \sum_{t=1}^{T}\Delta^t + \max\left\{2\sqrt{\frac{2L^2+d}{d}}\sqrt{\sum_{t=q}^{T}\mathbb{E}[\Delta^t]}, 6L^2\left(\frac{1+d}{d}\right)\sqrt{\ln(1/\gamma)}\right\}\sqrt{\ln(1/\gamma)}$$

$$(41)$$

We can further bound $\sum_{t=1}^{T}\mathbb{E}[\Delta^t]$ by using the fact that (41) is in the form of $s^2 \leq r + \max\{2as, 6bc\}c$ which can be solved for $s$ using quadratic formula. We solve this quadratic in Section A.4.2, and show that it implies $s \leq r + 2ac\sqrt{r} + \max\{4a^2, 6b\}c^2$. Using the solution of the quadratic formula, we get the following final result describing the effect of bandit feedback. With probability $1 - 4\gamma\ln(T)$,

$$\sum_{t=1}^{T}\mathbb{E}[\Delta^t] \leq \sum_{t=1}^{T}\Delta^t + 2\sqrt{\frac{2L^2+d}{d}}\sqrt{\sum_{t=1}^{T}\Delta^t}\sqrt{\ln(1/\gamma)} + \max\left\{\frac{8L^2+4d}{d}, 6L^2\left(\frac{1+d}{d}\right)\right\}\ln(1/\gamma)$$

$$(42)$$

In summary, we bound the difference between the effect of the bias $\sum_{t=1}^{T} \mathbb{E}[\Delta^t]$ and the empirical loss we minimize $\sum_{t=1}^{T} \Delta^t$ without the noise term. In the next section, we procede to bound the empirical loss $\sum_{t=1}^{T} \mathcal{L}^t$ and the effect of the noise term $\sum_{t=1}^{T} |\Delta^t - \mathcal{L}^t|$.

### A.3.3 ANALYSIS OF THE EMPIRICAL LOSS

In this section, we will analyze the empirical loss. Our analysis is similar to the regret analysis of Follow the Regularized Leader (FTRL). However, we do not get an adverserial bound. Our resulting bound is the function of distances of iterates denoted as $\|\mathbf{x}^t - \mathbf{x}^{t-1}\|_2$. Such a bound would not be useful in adverserial setting since adversary chooses the iterates. However, we also design the optimization method. Hence, we bound $\|\mathbf{x}^t - \mathbf{x}^{t-1}\|_2$ by setting step sizes accordingly.

We start our analysis with bounding the total empirical loss in terms of the length of the trajectory the learner takes. As a consequence of the FTL-BTL Lemma (Kalai & Vempala, 2005),

$$\sum_{t=1}^{T} \mathcal{L}(\mathbf{s}^t, \mathbf{x}^t, \xi^t, \theta^t, \psi^t) \leq \sum_{t=1}^{T} \left[ \mathcal{L}(\mathbf{s}^t, \mathbf{x}^t, \xi^t, \theta^t, \psi^t) - \mathcal{L}(\mathbf{s}^t, \mathbf{x}^t, \xi^t, \theta^{t+1}, \psi^{t+1}) \right] + \mathcal{R}(\theta^T, \psi^T).$$

$$(43)$$

We use the Lipschitz smoothness property to convert this into distance travelled by the learner as

$$\mathcal{L}(\mathbf{s}^t, \mathbf{x}^t, \xi^t, \theta^t, \psi^t) - \mathcal{L}(\mathbf{s}^t, \mathbf{x}^t, \xi^t, \theta^{t+1}, \psi^{t+1})$$

$$= \left( \frac{y(\mathbf{x}^t, \mathbf{s}^t, \xi^t)}{2\delta} - \mathbf{s}^{t\mathsf{T}} \nabla_{\mathbf{x}} g(r(\mathbf{x}^t; \theta^t); \psi^t) \right)^2 - \left( \frac{y(\mathbf{x}^t, \mathbf{s}^t, \xi^t)}{2\delta} - \mathbf{s}^{t\mathsf{T}} \nabla_{\mathbf{x}} g(r(\mathbf{x}^t; \theta^{t+1}); \psi^{t+1}) \right)^2$$

$$\leq 4L \sum_{t=1}^{T} \mathbf{s}^{t\mathsf{T}} (\nabla_{\mathbf{x}} g(r(\mathbf{x}^t; \theta^t); \psi^t) - \nabla_{\mathbf{x}} g(r(\mathbf{x}^t; \theta^{t+1}); \psi^{t+1})) \tag{44}$$

$$\leq 4L \underbrace{\sum_{t=1}^{T} \|\nabla_{\mathbf{x}} g(r(\mathbf{x}^t; \theta^t); \psi^t) - \nabla_{\mathbf{x}} g(r(\mathbf{x}^t; \theta^{t+1}); \psi^{t+1})\|_2}_{\text{LEARNERPATHLENGTH}}$$

With properly chosen $\lambda$, our regularizer enforces the smallest possible update, in terms of learner path length, which is consistent with the current sampled directions. This is due to the representability assumption which guarantees that manifold can be fit perfectly using the parametric family. Hence, there is a solution with $\mathcal{L} = 0$. Considering the regularizer is the learner path length, with proper choice of $\lambda$, the FTRL will choose the shortest learner path length.

Among all choices of $\mathbf{s}^t$, $\mathbf{s}^t = \nabla_{\mathbf{x}} F(\mathbf{x}^t, \xi^t) / \|\nabla_{\mathbf{x}} F(\mathbf{x}^t, \xi^t)\|$ would result in the longest distance. Hence, we can bound the learner path distance of our empirical problem with the distances of this oracle problem. We denote the $\theta$ and $\psi$ found by this oracle problem as $\hat{\theta}, \hat{\psi}$. Formally, this upper bound leads to

$$\text{LEARNERPATHLENGTH} \leq \sum_{t=1}^{T} \|\nabla_{\mathbf{x}} g(r(\mathbf{x}^t; \hat{\theta}^t); \hat{\psi}^t) - \nabla_{\mathbf{x}} g(r(\mathbf{x}^t; \hat{\theta}^{t+1}); \hat{\psi}^{t+1})\|_2$$

$$\leq \sum_{t=1}^{T} \|\nabla_{\mathbf{x}} g(r(\mathbf{x}^t; \hat{\theta}^t); \hat{\psi}^t) - \nabla_{\mathbf{x}} g(r(\mathbf{x}^{t-1}; \hat{\theta}^{t+1}); \hat{\psi}^{t+1})\|_2$$

$$+ \sum_{t=1}^{T} \|\nabla_{\mathbf{x}} g(r(\mathbf{x}^{t-1}; \hat{\theta}^{t+1}); \hat{\psi}^{t+1}) - \nabla_{\mathbf{x}} g(r(\mathbf{x}^t; \hat{\theta}^{t+1}); \hat{\psi}^{t+1})\|_2$$

$$(45)$$

$$\overset{(a)}{\leq} \sum_{t=1}^{T} \|\nabla_{\mathbf{x}} g(r(\mathbf{x}^t; \hat{\theta}^t); \hat{\psi}^t) - \nabla_{\mathbf{x}} g(r(\mathbf{x}^{t-1}; \hat{\theta}^t); \hat{\psi}^t)\|_2$$

$$+ \sum_{t=1}^{T} \|\nabla_{\mathbf{x}} g(r(\mathbf{x}^{t-1}; \hat{\theta}^{t+1}); \hat{\psi}^{t+1}) - \nabla_{\mathbf{x}} g(r(\mathbf{x}^t; \hat{\theta}^{t+1}); \hat{\psi}^{t+1})\|_2$$

$$\overset{(b)}{\leq} 2\mu \sum_{t=1}^{T} \|\mathbf{x}^t - \mathbf{x}^{t-1}\|_2,$$

as the consequence of the fact that oracle problem solves all gradients perfectly (i.e. $\nabla_{\mathbf{x}}\tilde{F}(\mathbf{x}^t, \xi^t) = \nabla_{\mathbf{x}}g(r(\mathbf{x}; \theta^i); \psi^i)$ for all $i > t$) in $(a)$, and the functions are $\mu$−smooth in $(b)$. Using the fact that gradient norms are bounded,

$$\sum_{t=1}^{T} \mathcal{L}(\mathbf{s}^t, \mathbf{x}^t, \xi^t, \theta^t, \psi^t) \leq 8\mu L \sum_{t=1}^{T} \|\mathbf{x}^t - \mathbf{x}^{t-1}\|_2 + 2L. \tag{46}$$

In order to extend this result to the $\Delta^t$, we use the smoothness of the function as

$$\begin{aligned} \Delta^t &= \left(\mathbf{s}^{t\mathsf{T}}[\nabla_{\mathbf{x}}\hat{F}(\mathbf{x}^t, \xi) - \nabla_{\mathbf{x}}g(r(\mathbf{x}^t; \theta^t); \psi^t)]\right)^2 \\ &\leq \left(\mathbf{s}^{t\mathsf{T}}\nabla_{\mathbf{x}}\hat{F}(\mathbf{x}^t, \xi) - \frac{y(\mathbf{x}^t, \mathbf{s}^t, \xi^t)}{2\delta}\right)^2 + \left(\frac{y(\mathbf{x}^t, \mathbf{s}^t, \xi^t)}{2\delta} - \mathbf{s}^{t\mathsf{T}}\nabla_{\mathbf{x}}g(r(\mathbf{x}^t; \theta^t); \psi^t)\right)^2 \\ &\leq \left(\mathbb{E}_{\mathbf{v}\in\mathbb{B}^d}\left[\mathbf{s}^{t\mathsf{T}}\nabla_{\mathbf{x}}F(\mathbf{x}^t + \delta\mathbf{v}, \xi) - \frac{y(\mathbf{x}^t, \mathbf{s}^t, \xi^t)}{2\delta}\right]\right)^2 + \mathcal{L}(\mathbf{s}^t, \mathbf{x}^t, \xi^t, \theta^t, \psi^t) \\ &\leq \mu^2\delta^2 + \mathcal{L}(\mathbf{s}^t, \mathbf{x}^t, \xi^t, \theta^t, \psi^t) \end{aligned} \tag{47}$$

By combining (47) and (46), we state

$$\sum_{t=1}^{T} \Delta^t \leq 8\mu L \sum_{t=1}^{T} \|\mathbf{x}^t - \mathbf{x}^{t-1}\|_2 + \mu^2\delta^2 T + 2L. \tag{48}$$

### A.3.4 PROOF OF THE THEOREM

We combine the aforementioned three arguments to state the final sample complexity of our method. Our analysis of SGD with bias from (36) combined with the definition of the $\Delta^t$ gives the following bound on the sample complexity.

$$\frac{1}{T}\sum_{t=1}^{T} \|\nabla_{\mathbf{x}}f(\mathbf{x}^t)\|_2^2 \leq \frac{\Omega}{\alpha T} + \frac{\mu\alpha\mathbf{V_g}}{2} + \frac{\Omega}{T}\sum_{t=1}^{T}\sqrt{d\mathbb{E}[\Delta^t]} + \delta^2\mu^2. \tag{49}$$

Using the concavity of the square root function with Jensen's inequality, we can convert this bound to

$$\frac{1}{T}\sum_{t=1}^{T} \|\nabla_{\mathbf{x}}f(\mathbf{x}^t)\|_2^2 \leq \frac{\Omega}{\alpha T} + \frac{\mu\alpha\mathbf{V_g}}{2} + \Omega\sqrt{d}\sqrt{\frac{1}{T}\sum_{t=1}^{T}\mathbb{E}[\Delta^t]} + \delta^2\mu^2. \tag{50}$$

Next, we will bound $\sqrt{\frac{1}{T}\sum_{t=1}^{T}\mathbb{E}[\Delta^t]}$ using (42). For simplicity, we will analyze two cases $(\sum_{t=1}^{T}\Delta^t \leq 1)$ and $(\sum_{t=1}^{T}\Delta^t > 1)$ seperately.

**Case 1,** $\sum_{t=1}^{T}\Delta^t \leq 1$**:** We substitute this bound directly in (42). With probability $1 - 4\gamma\ln(T)$,

$$\frac{1}{T}\sum_{t=1}^{T}\mathbb{E}[\Delta^t] \leq \frac{1}{T} + \frac{2}{T}\sqrt{\frac{2L^2 + d}{d}}\sqrt{\ln(1/\gamma)} + \max\left\{\frac{8L^2 + 4d}{dT}, 6L^2\left(\frac{1+d}{dT}\right)\right\}\ln(1/\gamma) \tag{51}$$

Relaxing the upper bound with the fact that dimension is greater than 1,

$$\sqrt{\frac{1}{T}\sum_{t=1}^{T}\mathbb{E}[\Delta^t]} \leq \sqrt{\frac{c_1}{T}} \tag{52}$$

where $c_1 = 1 + \sqrt{(8L^2 + 4)\ln(1/\gamma)} + \max\{8L^2 + 4, 12L^2\}\ln(1/\gamma)$.

**Case 2,** $\frac{1}{T}\sum_{t=1}^{T}\Delta^t > 1$**:** Using the fact that $\sqrt{x} < x$ for $x > 1$ and $\sqrt{x + y} \leq \sqrt{x} + \sqrt{y}$ as well as $d \geq 1$, we can state that with probability $1 - 4\gamma\ln(T)$,

$$\sqrt{\frac{1}{T}\sum_{t=1}^{T}\mathbb{E}[\Delta^t]} \leq \sqrt{\frac{c_2}{T}} + c_3\sqrt{\frac{1}{T}\sum_{t=1}^{T}\Delta^t} \tag{53}$$

where $c_2 = \max\{8L^2 + 4, 12L^2\} \ln(1/\gamma)$ and $c_3 = 1 + \sqrt{2\sqrt{(2L^2+1)\ln(1/\gamma)}}$. Combining this with the bound (48) and $\sqrt{x} \leq x$ for $x > 1$,

$$\sqrt{\frac{1}{T}\sum_{t=1}^{T}\mathbb{E}[\Delta^t]} \leq \sqrt{\frac{c_2}{T}} + \frac{2c_3 L}{T} + c_3(\mu^2\delta^2 + 8\mu\alpha\mathbf{V_g}) \tag{54}$$

We combine two cases and substitute it in (50). The final sample complexity is,

$$\frac{1}{T}\sum_{t=1}^{T}\|\nabla_{\mathbf{x}}f(\mathbf{x}^t)\|^2 \leq \frac{\Omega}{\alpha T} + \mu\alpha\mathbf{V_g}\left(\frac{1}{2} + 8c_3\Omega\sqrt{d}\right) + \mu^2\delta^2\left(1 + \Omega c_3\sqrt{d}\right) + \frac{2c_3\Omega L\sqrt{d}}{T} + \sqrt{\frac{c_{1,2}d}{T}}. \tag{55}$$

where $c_{1,2} = \sqrt{\Omega}\max\{c_1, c_2\}$. We minimize with respect to $\alpha^t$ and substitute it.

$$\frac{1}{T}\sum_{t=1}^{T}\|\nabla_{\mathbf{x}}f(\mathbf{x}^t)\|^2 \leq \sqrt{\frac{2\mu\mathbf{V_g}\Omega}{T}}\left(1 + \sqrt{4c_3\Omega d}\right) + \mu^2\delta^2\left(1 + \Omega c_3\sqrt{d}\right) + \frac{2c_3\Omega L\sqrt{d}}{T} + \sqrt{\frac{c_{1,2}d}{T}}. \tag{56}$$

Before we solve for $\delta$, we bound $\mathbf{V_g}$ by choosing $\beta = 1/d$ as,

$$\mathbb{E}[\mathbf{V_g}] \leq \mathbb{E}\left[\left(\frac{1}{d}\mathbf{g}_e + \left(1 - \frac{1}{d}\right)\mathbf{g}_m\right)\right] \leq 4L^2 n^2 + \frac{4n^2 V_F}{\delta^2}. \tag{57}$$

Next, we solve $\delta$ to obtain the statement of the theorem. With probability $1 - 4\gamma\ln(T)$,

$$\frac{1}{T}\sum_{t=1}^{T}\|\nabla_{\mathbf{x}}f(\mathbf{x}^t)\|^2 \leq \frac{k_1 d^{\frac{1}{2}}}{T} + \frac{k_2 d^{\frac{1}{2}} + k_3 n + k_4 n d^{\frac{1}{2}}}{T^{\frac{1}{2}}} + \frac{k_5 n^{\frac{2}{3}} + k_6 d^{\frac{1}{2}} n^{\frac{2}{3}}}{T^{\frac{1}{3}}} \tag{58}$$

where $k_1 = 2c_3\Omega L$, $k_2 = \sqrt{c_{1,2}}$, $k_3 = 2L\sqrt{2\mu\Omega}$, $k_4 = 4L\Omega\sqrt{\mu c_3}$, $k_5 = 3(2\Omega V_F)^{1/3}$, and $k_6 = k_5(3\Omega c_3)$. $\qquad\square$

## A.4 USEFUL ELEMENTARY RESULTS

### A.4.1 EXPECTATION OF $(\mathbf{s}^\intercal\mathbf{v})^2$ WHEN $\mathbf{s}$ IS CHOSEN UNIFORMLY FROM $\mathbb{S}^{d-1}$

Consider $\int F(\mathbf{Oe})\mu(\mathbf{O})$ where $\int[\cdot]\mu(\mathbf{O})$ is an integral over orthogonal matrices with Haar measure. If $\mathbf{e}$ is a unit vector, we can show that

$$\mathbb{E}_{\mathbf{s}\in\mathbb{S}^{d-1}}[F(\mathbf{s})] = \int F(\mathbf{Oe})\mu(\mathbf{O}). \tag{59}$$

Before we use this result, we define $\|\mathbf{v}\|^2$ as an integral over orthogonal matrices $\mathbf{O}$. Using orthogonality and cyclic property of the trace,

$$\mathbf{v}^\intercal\mathbf{v} = \text{Tr}(\mathbf{vv}^\intercal) = \int \text{Tr}(\mathbf{vv}^\intercal)\mu(\mathbf{O}) = \int \text{Tr}\left(\mathbf{OO}^\intercal\mathbf{vv}^\intercal\right)\mu(\mathbf{O}) = \int \text{Tr}\left(\mathbf{O}^\intercal\mathbf{vv}^\intercal\mathbf{O}\right)\mu(\mathbf{O}). \tag{60}$$

Since the indentity matrix is sum of outer products of one hot vectors $\mathbf{e}_i$ as $\mathbf{I} = \sum_i \mathbf{e}_i\mathbf{e}_i^\intercal$,

$$\int \text{Tr}\left(\mathbf{O}^\intercal\mathbf{vv}^\intercal\mathbf{O}\right)\mu(\mathbf{O}) = \int \text{Tr}\left(\sum_i\mathbf{e}_i\mathbf{e}_i^\intercal\mathbf{O}^\intercal\mathbf{vv}^\intercal\mathbf{O}\right)\mu(\mathbf{O}) = \sum_i\int \text{Tr}\left(\mathbf{e}_i^\intercal\mathbf{O}^\intercal\mathbf{vv}^\intercal\mathbf{Oe}_i\right)\mu(\mathbf{O}). \tag{61}$$

Using (59), we can further show

$$\sum_i\int \text{Tr}\left(\mathbf{e}_i^\intercal\mathbf{O}^\intercal\mathbf{vv}^\intercal\mathbf{Oe}_i\right)\mu(\mathbf{O}) = \sum_i\mathbb{E}_{\mathbf{s}\in\mathbb{S}^{d-1}}[\text{Tr}(\mathbf{s}^\intercal\mathbf{vv}^\intercal\mathbf{s})] = d\mathbb{E}_{\mathbf{s}\in\mathbb{S}^{d-1}}[(\mathbf{s}^\intercal\mathbf{v})] \tag{62}$$

Hence, combining all,

$$\mathbb{E}_{\mathbf{s}\in\mathbb{S}^{d-1}}[(\mathbf{s}^\intercal\mathbf{v})] = \frac{\|\mathbf{v}\|^2}{d}. \tag{63}$$

A.4.2 Bounding the quadratic form

We need to bound $s$ when the following quadratic inequality is correct;

$$s \le r + \max\{2a\sqrt{s}, 6bc\}c. \tag{64}$$

We first consider the max as two separate options. Given the original inequality, one of the following is correct;

$$s \le r + 2ac\sqrt{s}, \qquad or \qquad s \le r + 6bc^2. \tag{65}$$

For the first case, $(\sqrt{s})^2 - 2ac\sqrt{s} - r \le 0$. Using the quadratic formula, $\sqrt{s}$ is smaller than the largest root as $\sqrt{s} \le ac + \sqrt{a^2c^2 + r}$. Hence,

$$s = (\sqrt{s})^2 \le (ac + \sqrt{a^2c^2 + r})^2 = a^2c^2 + a^2c^2 + r + 2ac\sqrt{a^2c^2 + r} \le 4a^2c^2 + 2ac\sqrt{r} + r \tag{66}$$

Combining the resulting bound with the other option gives the final bound as,

$$s \le r + 2ac\sqrt{r} + \max\{4a^2, 6b\}c^2 \tag{67}$$

# B Additional Implementation Details

In this section, we provide additional details for the implementation. Our experimental setup is also available open-source along with a full implementation of our method at `https://github.com/intel-isl/LMRS`. One key element of our method is the parametric family we use to learn the manifold. We consider a multi-layered perceptron with one to three hidden layers as $\text{Linear}(d, 2n) \to \text{ReLU} \to \text{Linear}(2n, n) \to \text{ReLU} \to \text{Linear}(n, n)$ for $d < 1000$ and $\text{Linear}(d, 1/2d) \to \text{ReLU} \to \text{Linear}(1/2d, 2n) \to \text{ReLU} \to \text{Linear}(2n, n) \to \text{ReLU} \to \text{Linear}(n, n)$ for $d > 1000$. We set the dimensionality of the manifold as the number of directions $k$ which is a hyper parameter. We use grid search over $\delta$ and $n = k$ values and choose the best performing one in all experiments. Moreover, we also reinitialize the manifold parameters whenever the estimated gradient's magnitude is less than $1e-6$. We perform online gradient descent to learn the model parameters using SGD with momentum as $0.9$. We also perform grid search for learning rate over $\{1e-4, 1e-3, 1e-2\}$. We set $\lambda = 10^3$ for all experiments.

We further discuss experiment-specific details below.

**MuJoCo experiments.** We use linear policies and initialize them as zeros, which corresponds to no action. We use $v2-t$ algorithm from Mania et al. (2018), which includes whitening of the observation space and using top-k directions instead of all. We use grid search over the parameter space described by Mania et al. (2018) for $n = k$, $\alpha$ and $\delta$.

**Low-dimensional unconstrained optimization suite.** We use the following functions: *sphere, noisysphere, cigar, tablet, cigtab, cigtab2, elli, rosen, rosen_chained, diffpow, rosenelli, ridge, ridge-circle, happycat, branin, goldsteinprice, rastrigin, schaffer, schwefel2_22, lincon, rosen_nesterov, styblinski_tang, bukin* with dimensions $d = 10$ and $d = 100$ resulting in total 46 problems. We initialize all solutions with zero mean unit variance normal variables and use grid search over $\delta \in \{1e-4, 1e-3, 1e-2, 1e-1\}$, $k \in \{2, 5, 10, 50\}$, and $\alpha \in \{1e-4, 1e-3, 1e-2, 1e-1\}$.

**Airfoil optimization.** We initialize the parameters of the manifold with zero-mean unit-variance normal variables. We use grid search over $\delta \in \{1e-4, 1e-3, 1e-2, 1e-1\}$, $k \in \{2, 5, 10, 50\}$, and $\alpha \in \{1e-4, 1e-3, 1e-2, 1e-1\}$. For choosing hyper-parameters, we simulate all models with Reynolds number $12e6$, speed $0.4$ mach, and angle of attack 5 degrees. After the hyper-parameters are set, we used Reynolds number $14e6$, speed $0.6$ mach and angle of attack 2 degrees for evaluation.

**Implementation details for the baselines.** For ARS, we use grid search over the parameters recommended by Mania et al. (2018). For CMA-ES, we use pycma (Hansen et al., 2019) with recommended hyperparameters. For Bayesian optimization, we use the REMBO optimizer (Wang et al., 2016) with the following details. For continuous control problems, we use Structured Kernel Interpolation (SKI) (Wilson & Nickisch, 2015) and perform grid search over hyperparameters using the ranges recommended by GPyTorch (Gardner et al., 2018). For other problems, we use full GP inference since the number of samples is rather low and perform grid search over kernel parameters following GPyTorch (Gardner et al., 2018). As an acquisition function, we perform grid search over Optimistic Expected Improvement (Rontsis et al., 2017), Multi-point Expected Improvement (Marmin et al., 2015), and the approach of Ginsbourger et al. (2010).

