# OpenReview forum: "Learning to Guide Random Search"
_ICLR.cc/2020/Conference — Accept (Poster)_

### Official Review · AnonReviewer2 · 2019-10-23
**Official Blind Review #2**

**Rating:** 6

**Review:**

Contributions:
	-Authors have proposed a methodology to optimise high dimensional functions in a derivative-free setup by reducing the sample complexity by simultaneously learning and optimising the low dimensional manifolds for the given high dimensional problem.

Although, performing dimensionality reduction to learn the low dimensional manifolds is popular in the research community, the extensions made and the approach authors have considered seems to be novel.

Comments:

	- Authors have talked about the utilization of domain knowledge on the geometry of the problem. How feasible is to expect the availability of the domain knowledge? Authors have not discussed the downsides of the proposed method if the domain knowledge is not available, and a possible strategy to overcome the same.
	- Authors have said that they are specifically interested in random search methods. Is there any motivating reason to stick to the random search methods? Why not consider other sample efficient search methods?
	“…….random search scale linearly with the dimensions”, why one should not consider other sample efficient methods that grow sub-linearly as against random search?
Srinivas, N., Krause, A., Kakade, S. M., and Seeger, M. Gaussian process optimization in the bandit setting: No regret and experimental design. International Conference on Machine Learning, 2010
	- Please derive Lemma 1 in the appendix for the sake of completeness.
	- I am missing a discussion about manifold parameters like “λ” in the important equations.
	- Authors have made a strong claim that neural networks can easily fit any training data, but it may be not be true for many datasets.
	 - Authors have claimed that they have fast and no-regret learning by selecting mixing weight β=1/d. Author might want to discuss more on this as this is an important metric.
	 - “ …. total time spent on learning the manifold is negligible…. ” – any supporting results for this claim.
	 - “…….communication cost from d+2k to d+2k+kd… ” – curious to know if there is any metric like wall-clock time to talk about the optimisation time.

	- Authors have restricted the comparisons to only three important methods, but it is always comparing with other baselines in the same line. Authors should consider Bayesian optimisation as it is a good candidate for the performance comparison, even though the researchers are interested only in random search methods (just like CMA–ES).
Kirschner, Johannes, Mojmír Mutný, Nicole Hiller, Rasmus Ischebeck, and Andreas Krause. "Adaptive and Safe Bayesian Optimization in High Dimensions via One-Dimensional Subspaces." arXiv preprint arXiv:1902.03229 (2019).

	- It is seen from the results that the proposed method is not performing better for low dimensional problem like “Swimmer” function. But according to the initial claim, method was supposed to work better in low dimensional problems. Is it because of the fact that the problem space is not drawn from high dimensional data distributions?
	- “…..improvement is significant for high dimensional problems” – It will be better if the authors compare their proposed method with some more derivative-free optimisers that are proven to be good in high dimensions (like high dimension Bayesian optimisation).
	-  “The no-learning baseline outperforms random search ……….” – this statement is not very clear, does it mean like the proposed method works only when the problem is reduced from higher dimensions to lower dimensions and not on the lower dimensional problem itself?
	- “Performance profiles represent how frequently a method is within the distance T of optimality” – Any thumb rule considered for the choice of T?. Can we think of any relation with standard metrics like simple regret or cumulative regret that are used to measure the optimisation performance?
	- “Although BO methods typically do not scale…… ” – Authors have made a strong assumption here. In the literature, we see active research in the context of high dimensional optimisation.
Rana, Santu, Cheng Li, Sunil Gupta, Vu Nguyen, and Svetha Venkatesh. "High dimensional Bayesian optimization with elastic gaussian process." In Proceedings of the 34th International Conference on Machine Learning-Volume 70, pp. 2883-2891. JMLR. org, 2017.



Minor Issues:
	“Improve the sample complexity” may not convey the meaning very clearly to the readers, something like “Improve the sample efficiency” or “Reduce the sample complexity” would add more clarity.
	Inconsistency in the terms used in Proposition 1 and Proposition 2. What does “I” signify in the formula? Was that supposed to be “t”?
	Even though the constants are mentioned in the appendix, it is always better to mention the constants used in the algorithm like “α“ as step size for quick understanding.
	“Follow The Regularized Leader (FTRL)” is more appropriate than “follow the regularized leader (FTRL)”
	“noise table in pre-processing” – Should it mean something relevant to the paper?
	“We use widely used…..” – may be consider rephrasing the sentence here
	“treshold” – Typo in Table 1
	Y – Axis in Figure 2 is missing
	Appendix B :
	“We also perform grid search …. “ would look better
	MuJoCo Experiments – is the parameter space continuous and what is the search space considered for n, α and δ. Do we deal with smaller search spaces in every problem? Any other way of searching the parameter space to further improve the efficiency?


**Experience Assessment:**

I have published one or two papers in this area.

**Review Assessment: Checking Correctness Of Derivations And Theory:**

I assessed the sensibility of the derivations and theory.

**Review Assessment: Checking Correctness Of Experiments:**

I assessed the sensibility of the experiments.

**Review Assessment: Thoroughness In Paper Reading:**

I read the paper at least twice and used my best judgement in assessing the paper.

---

> ### Author Response · Authors · 2019-11-13
> **Response to Review #2**
>
> We thank the reviewer for the time and effort, as well as the encouraging comments. We address the concerns as follows.
>
> Q: “How feasible is to expect the availability of the domain knowledge? ... downsides of the proposed method if the domain knowledge is not available, and a possible strategy to overcome the same.”
>
> A: The specific form of domain knowledge we need is an architectural specification and the fact that the problem lies in some low-dimensional manifold. If any of the deep learning/representation learning methods have already been utilized for the domain, this domain knowledge typically already exist. We believe there are a large number of such domains for our method to be impactful. If such knowledge does not exist, we expect our method to perform the same as the original random search.
>
>
> Q: Is there any motivating reason to stick to the random search methods?
>
> A: The main reason for sticking to random search is existing literature (e.g., Mania 2018, Vemula 2019) showing their practicality for the problems of our interest. For example, they perform well for model-free learning of continuous control of a robot. Moreover, our additional Bayesian optimization results suggest that applying Bayesian optimization to the problems of interest is not straightforward.
>
>
> Q: “ …. total time spent on learning the manifold is negligible…. ” – any supporting results for this claim.  “…….communication cost from d+2k to d+2k+kd… ” – curious to know if there is any metric like wall-clock time...
>
> A: We added a study on the wall-clock computation times in Fig 2 (see Section 5.1 for discussion).
>
>
> Q: “ Authors should consider Bayesian optimisation as it is a good candidate for the performance comparison” and “It will be better if the authors compare their proposed method with some more derivative-free optimisers ... (like high dimension Bayesian optimisation).”
>
> A: We added an additional comparison to Bayesian optimization. We chose the REMBO optimizer as it can scale to very high-dimensional spaces. Our results suggest that Bayesian optimization fails for continuous control problems on MuJoCo, and we discuss possible reasons in Section 5.1. For other problems, our algorithm either outperforms BO or performs similarly. Moreover, BO outperforms all other local search heuristics (CMA-ES, GuidedES, ARS) for these problems. See Section 5 for more details.
>
>
> Q: “Performance profiles represent how frequently a method is within the distance Tau of optimality” – Any thumb rule considered for the choice of Tau?. Can we think of any relation with standard metrics like simple regret or cumulative regret that are used to measure the optimisation performance?
>
> A: We reworded the explanation of performance profiles as it was somewhat confusing. We do not choose the Tau. It is the x-axis of the performance profile. Intuitively, Tau represents the difference between the sample complexity of the method and the best performing one (the one with the lowest sample complexity).
>
>
> Q: “Although BO methods typically do not scale…… ” Authors have made a strong assumption here.
>
> A: We removed this sentence entirely.
>
>
> Q: It is seen from the results that the proposed method is not performing better for low dimensional problem like “Swimmer” function. But according to the initial claim, method was supposed to work better in low dimensional problems. Is it because of the fact that the problem space is not drawn from high dimensional data distributions?
>
> A: Our method is best suited for high-dimensional problems lying in a low-dimensional manifold. For low-dimensional problems (like Swimmer), random search in the full space already performs well enough.
>
>
> Q: “The no-learning baseline outperforms random search ……….” – this statement is not very clear…”, does it mean like the proposed method works only when the problem is reduced from higher dimensions to lower dimensions and not on the lower dimensional problem itself?
>
> A: We reworded the discussion. This result means that for high-dimensional problems, online learning is crucial. The no-learning baseline uses random features instead of our proposed method.
>
>
> Q: Please derive Lemma 1 in the appendix for the sake of completeness
>
> A: We are not sure which Lemma the reviewer is referring to? We already proved Lemma 1 in Appendix C.1.
>
>
> MINOR QUESTIONS:
>
> Q: “what is the search space considered for n, α and δ.”:
>
> A: Although the search spaces for α and δ are continuous, we use discrete values. We already state the search spaces in Appendix B.
>
>
> Q:Do we deal with smaller search spaces in every problem? Any other way of searching the parameter space to further improve the efficiency?:
>
> A:We already have ideas on how to incorporate ideas from Bayesian optimization and/or Hyperband into our method. They are not straightforward and we consider them for future work.
>
> REMARK:
> The i in Proposition 1&2 is a typo, it is supposed to be t. We fixed this and all other typos pointed out by the reviewer.

---

### Official Review · AnonReviewer3 · 2019-10-23
**Official Blind Review #3**

**Rating:** 6

**Review:**

The computation times for random search methods depend largely on the total dimension of the problem. The larger the problem, the longer it takes to perform a single iteration.  I believe the main reason why many people use deep reinforcement learning to solve their problems is due to its dimension-independence.  I am not aware of a paper that tries to minimize the sample complexity. Thus, I think the idea in this paper is novel and may have influence on the literature (maybe an encouragement for a shift from deep reinforcement learning to derivative-free optimization methods).

In terms of presented results I think that there is not much that they could do wrong. They show in Figure 1 that the reward they achieved with their method is only outperformed by Augmented Random Search (ARS) on the Ant task. On all other tasks, their method at least performs on par with ARS which is a good result.

In Table 1 they show the number of episodes that are needed to achieve the reward threshold. Their method required less episodes than all other methods, but I think this is not the only criteria they should have looked at. So, it might be the case that their iterations take longer to compute than the iterations of the ARS and thereby making it slower.

The authors have showed that their method has a lower sample complexity, which is their goal of the research (“Our major objective is to improve the sample efficiency of random search.”). However, I am not sure whether this means that it also has a lower computational complexity. They address this issue briefly by stating that “Our method increases the amount of computation since we need to learn a model while performing the optimization. However, in DFO, the major computational bottleneck is typically the function evaluation. When efficiently implemented on a GPU, total time spent on learning the manifold is negligible in comparison to function evaluations.” This would mean that their iterations are performed in less computation time than the ARS, but I would have personally liked to see a number attached to this.
If we thus assume that this is the case, then their results are sound. However, I do not see this reduced complexity reflected in the results. If I look at the ratios between the number of episodes it takes to solve the tasks, they seem to be similar to the ones from the ARS. The number of episodes reduces by roughly 50% for all tasks but this keeps the ratio between the different tasks identical. I would have assumed that the ratios would increase in the favor of the larger problems like the Humanoid task. In other words, I still see the influence of the larger dimension in the results. Maybe I am too critical, but to me if they would have just found a faster method without the reduced sample complexity, they would have achieved similar results.

Of course, this problem would not be present if the computation time increases with the number of iterations. In that case, the computation time would not reduce by a “fixed” ratio and would therefore decrease relatively much on the tasks with a higher dimension. But that would require an exact comparison between the computation times for all tasks for both their method and the ARS which I do not see in their results. If all these things are common knowledge, then their results are sound and they have found a large improvement to the already well performing ARS.


**Experience Assessment:**

I have read many papers in this area.

**Review Assessment: Checking Correctness Of Derivations And Theory:**

I assessed the sensibility of the derivations and theory.

**Review Assessment: Checking Correctness Of Experiments:**

I carefully checked the experiments.

**Review Assessment: Thoroughness In Paper Reading:**

I read the paper at least twice and used my best judgement in assessing the paper.

---

> ### Author Response · Authors · 2019-11-13
> **Response to Review#3**
>
> We thank the reviewer for the time and effort, as well as the encouraging comments. We address the concerns as follows:
>
> Q: “This would mean that their iterations are performed in less computation time than the ARS, but I would have personally liked to see a number attached to this.”
>
> A: We added a wall-clock time analysis in Section 5.1. The results and conclusions are the same: our method outperforms all other baselines in wall-clock time as well. Moreover, the computation of all methods is negligible when compared with simulation time. Hence, the shapes of the curves are also similar to the sample complexity case.
>
> Q: “...The number of episodes reduces by roughly 50% for all tasks but this keeps the ratio between the different tasks identical. I would have assumed that the ratios would increase in favor of the larger problems like the Humanoid task”
>
> A: We thank the reviewer for this interesting analysis and the comment. The ratio of improvement is between 1.7 and 3.7 times, and the trend seems not exactly to follow dimensionality. It is important to note that both the problem dimension ($d$) and manifold dimension ($n$) are changing between each experiment. Hence, we believe it is not easy to make any conclusion from MuJoCo experiments. To understand this phenomenon in a more controlled environment, we designed synthetic problems with controllable manifold and problem dimensions and compared our method with the baseline random search. We include this study in Appendix A, and the effect of manifold dimensionality is very clear.

---

### Official Review · AnonReviewer1 · 2019-10-30
**Official Blind Review #1**

**Rating:** 6

**Review:**


In this paper, the authors first improve the gradient estimator in (Flaxman et al., 2004) zeroth-order optimization by exploiting low-rank structure. Then, the authors exploit machine learning to automatically discover the lower dimensional space in which the optimization is actually conducted. The authors justified the proposed algorithm both theoretically and empirically. The empirical performances of the proposed estimator outperforms the current derivative-free optimization algorithms on MuJoCo for policy optimization.

The paper is well-motivated and well-organized. I really like this paper, which provide an practical algorithm with theoretical guarantees (although under some mild conditions). The empirical comparison also looks promising, for both RL problems and zeroth-order optimization benchmark.

I have roughly checked the proofs. The main body of the proof looks reasonable to me. However, I have some questions about one detail: In the proof of lemma 1, how the forth equation comes form third equation is not clear. Only manifold stokes' theorem might not enough since there is Us in side of f while U^*s outside of f. I think there should be one more bias term.


For the empirical experiment, it is a pity that the algorithm is not compared with Bayesian optimization, which is also an important baseline.  I am expecting to see the performances comparison between these two kinds of algorithms.

Minor:

The "unbiasedness" of the gradient should be more clear. It is NOT unbiased gradient w.r.t. the original function, but the smoothed version.

=====================================================================

Thanks for the reply. The comparison between the proposed algorithm and BO looks promising.

I will keep my score.

I am still confused about the proof of lemma 1. Following the notations in the paper, I was wondering the unbiased gradient should be

$E_{S}[f(x + \delta U^*s)U^*s]$

Then, the lemma should characterize the difference between

$E_{S}[f(x + \delta Us)Us]$ and $E_{S}[f(x + \delta U^*s)U^*s]$.

However, current lemma 1 is not bounding this error.


**Experience Assessment:**

I have read many papers in this area.

**Review Assessment: Checking Correctness Of Derivations And Theory:**

I assessed the sensibility of the derivations and theory.

**Review Assessment: Checking Correctness Of Experiments:**

I assessed the sensibility of the experiments.

**Review Assessment: Thoroughness In Paper Reading:**

I read the paper at least twice and used my best judgement in assessing the paper.

---

> ### Author Response · Authors · 2019-11-13
> **Response to Review#1**
>
> We thank the reviewer for the time and effort. We also appreciate the encouraging comments, and address the concerns as follows:
>
> Q: In the proof of lemma 1, how the fourth equation comes from the third equation is not clear.
>
> A: It follows from the orthonormality of the $U$ and $U^\star$. We added additional explicit steps (step b and d in Appendix C.1) to clarify this.
>
> Q: “...it is a pity that the algorithm is not compared with Bayesian optimization, which is also an important baseline…”
>
> A: We added an additional comparison to Bayesian optimization (BO). We chose the REMBO optimizer as it can scale to very high-dimensional spaces. Our results suggest that Bayesian optimization fails for continuous control problems on MuJoCo, and we discuss possible reasons in Section 5.1. For other problems, our algorithm either outperforms BO or performs similarly. Moreover, BO outperforms all other local search heuristics (CMA-ES, GuidedES, ARS) for these problems. See Section 5 for more details.
>
> Q: “The "unbiasedness" of the gradient should be more clear. It is NOT unbiased gradient w.r.t. the original function, but the smoothed version.”
>
> A: We went over the manuscript and carefully clarified/re-worded every time we used the word "unbiased."

---

### Official Review · AnonReviewer4 · 2019-11-04
**Official Blind Review #4**

**Rating:** 8

**Review:**

This paper addresses the problem of optimizing high dimensional functions lying in low dimensional manifolds in a derivative-free setting. The authors develop an online learning framework which jointly learns the nonlinear manifold and solves the optimization. Moreover, the authors present a bound on the convergence rate of their algorithm which improves the sample complexity upon vanilla random search. The paper is overall well written and the core idea seems interesting. However, the reviewer has a few concerns which needs to be addressed.

1) Methodology: This work depends on deep networks to learn the nonlinear manifolds which is justifiable by the power of deep nets. However, several issues may arise.

1.1)  Globally optimizing the loss function of a deep network is no easy task and according to the authors, their theoretical results holds only if equation (6)--which includes the loss function of a deep net-- is globally optimized.

1.2) Even if one could globally minimize the loss function up to a tolerance, this will require a large number of epochs resulting in a high overhead cost for each update of the algorithm. This cost should be considered during the evaluation of the performance of the algorithm.

1.3) Finally, although the authors mention that : " Experimental results suggest that neural networks can easily fit any training data", the success of neural networks highly depends on their architecture and carefully tuning their several hyperparameters including the number of hidden layers, the number of nodes in each such layer, the choice of activation function, the choice of optimization method, learning rate, momentum, dropout rate, data-augmentation parameters, etc. One evidence around the necessity of carefully tuning the neural networks lies in appendix B where the authors mention their specific choice of hyperparameters for each experiment as well as the cross validation range they have used. Again, the overhead cost of finding a good deep network through cross-validation or any other method of choice (such as Bayesian optimization or Hyperband) should be considered towards the total cost of the algorithm.

* Note that complex nonlinear manifolds might be better captured by complex yet flexible architecture as the authors also state that: "If the function of interest is known to be translation invariant, convolutional networks could be deployed to represent the underlying manifold structure". Hence, a simple fully connected network with fixed hyperparameters is suboptimal in capturing the different manifolds over various problems. This highlights the importance of exploring the space of hyperparameters.

2) Experiments: The results are reported solely over the number of episodes (function evaluations) while the cost of each episode might be significantly different among different methods. Thus, for a thorough examination, reporting the performance over wall-clock time is recommended and required, ideally in both serial and parallel settings . It does not matter whether the time is spent for a function evaluation or for reasoning about the manifold through training the deep network, it should be taken into account.

Minor issues:

1. On page 2, there is a typological error in the footnote in defining the L-Lipschitz concept (replace \mu with L).

2. On page 3, section 3.1, at the end of the second line, g should be a function of both \mathbf{r} and psi.

**Experience Assessment:**

I have published one or two papers in this area.

**Review Assessment: Checking Correctness Of Derivations And Theory:**

I assessed the sensibility of the derivations and theory.

**Review Assessment: Checking Correctness Of Experiments:**

I assessed the sensibility of the experiments.

**Review Assessment: Thoroughness In Paper Reading:**

I read the paper at least twice and used my best judgement in assessing the paper.

---

> ### Author Response · Authors · 2019-11-13
> **Response to Review#4**
>
> We thank the reviewers for their time and effort spent providing feedback. We address the concerns as follows:
>
> Q: 1.1 “Globally optimizing the loss function of a deep network is no easy task..”
>
> A: We agree that training a NN is no easy task. However, we clarify the following points: 1) Our theoretical results only need eq (6) to be minimized over the observed data points. In supervised learning terms, we only need training error to be low, which is a relatively easier task. 2) Our empirical results suggest that a simple optimizer (a few iterations of SGD) typically suffices for all the problems we tried. 3) The optimization theory of NNs is an active research area, and with a better understanding, our proposed method can further be improved in the future.
>
> Q: 1.2 “Even if one could globally minimize the loss function up to a tolerance, this will require a large number of epochs resulting in a high overhead cost”
>
> A: For practical implementation, we only performed very few iterations of SGD (less than 10) at each step. This suffices since we do not solve the optimization problem from scratch at every iteration, but rather start from the previous solution. We also added a study on the wall-clock computation times in Fig 2 (see Section 5.1 for discussion). Overhead of our method is negligible when compared with the simulation/function query time, which is typically the bottleneck in DFO.
>
> Q 1.3,1: “...success of neural networks highly depends on their architecture and carefully tuning their several hyperparameters including the number of hidden layers, the number of nodes in each such layer, the choice of activation function, the choice of optimization method, learning rate, momentum, dropout rate, data-augmentation parameters, etc. …overhead cost of finding a good deep network through cross-validation or any other method of choice (such as Bayesian optimization or Hyperband) should be considered towards the total cost of the algorithm.”
>
> A: We strongly agree with the reviewer on the importance of architectural specifications. However, we disagree that these choices need to be searched entirely. For many interesting domains (any domain in which deep learning has been applied to), some of these architectural choices are part of the domain knowledge. For example, we did NOT perform any search over architecture. Our architecture follows the common practices in the deep reinforcement learning domain as similar models are used in the literature.
>
>
> Q 1.3.2: “One evidence around the necessity of carefully tuning the neural networks lies in appendix B where the authors mention their specific choice of hyperparameters for each experiment as well as the cross validation range they have used…”
>
> A: In our empirical analysis, we tried to be as fair as possible by making search spaces of random search and our method as close as possible. To do this, we tie some hyper-parameters of our method to existing hyper-parameters. For example, the manifold dimension is set to be the same as the number of directions parameter in ARS (we denote this as $n=k$ in Appendix B). Moreover, we use a single regularizer multiplier ($\lambda$) in all experiments and keep the manifold learning rate the same. Quantitatively, the cardinality of the hyper-parameter space for ARS is 735 for MuJoCo and 256 for other problems. The cardinality of the hyper-parameter space for our method is 780 for MuJoCo and 256 for other problems. In summary, we believe our empirical comparison is fair.
>
> Q 2: “for a thorough examination, reporting the performance over wall-clock time is recommended and required, ideally in both serial and parallel settings”
>
> We also added a study on the wall-clock computation times in Fig 2 (see Section 5.1 for discussion). We only add parallel computation times as the serial search is not feasible (would require months of compute) for many of the problems we are interested in.
>
> Q: Minor issues:
> A: Thanks for pointing them out. We fixed them in the updated version.

---

### Decision · Program_Chairs · 2019-12-19

**Decision:**

Accept (Poster)

**Comment:**

This paper develops a methodology to perform global derivative-free optimization of high dimensional functions through random search on a lower dimensional manifold that is carefully learned with a neural network.  In thorough experiments on reinforcement learning tasks and a real world airfoil optimization task, the authors demonstrate the effectiveness of their method compared to strong baselines.  The reviewers unanimously agreed that the paper was above the bar for acceptance and thus the recommendation is to accept.  An interesting direction for future work might be to combine this methodology with REMBO.  REMBO seems competitive in the experiments (but maybe doesn't work as well early on since the model needs to learn the manifold).  Learning both the low dimensional manifold to do the optimization over and then performing a guided search through Bayesian optimization instead of a random strategy might get the best of both worlds?